# Function of bidirectional sensitivity in the otolith organs established by transcription factor Emx2

Young Rae Ji[1,7], Yosuke Tona [1,8], Talah Wafa [2], Matthew E. Christman[3], Edward D. Tourney[3], Tao Jiang[1,9], Sho Ohta [1], Hui Cheng[4], Tracy Fitzgerald[2], Bernd Fritzsch [5], Sherri M. Jones [6], Kathleen E. Cullen [3] & Doris K. Wu [1] ✉

Otolith organs of the inner ear are innervated by two parallel afferent projections to the brainstem and cerebellum. These innervations were proposed to segregate across the line of polarity reversal (LPR) within each otolith organ, which divides the organ into two regions of hair cells (HC) with opposite stereociliary orientation. The relationship and functional significance of these anatomical features are not known. Here, we show regional expression of Emx2 in otolith organs, which establishes LPR, mediates the neuronal segregation across LPR and constitutes the bidirectional sensitivity function. Conditional knockout (cKO) of *Emx2* in HCs lacks LPR. *Tmie* cKO, in which mechanotransduction was abolished selectively in HCs within the Emx2 expression domain also lacks bidirectional sensitivity. Analyses of both mutants indicate that LPR is specifically required for mice to swim comfortably and to traverse a balance beam efficiently, but LPR is not required for mice to stay on a rotating rod.

Falls are a leading cause of injury and death in the elderly population (Center for Disease Control and Prevention) and prevention treatments are limited. A better understanding of how vestibular information is being processed normally at the level of the inner ear and brain will help to design better strategies for prevention. The vestibular system of the mammalian inner ear consists of five major sensory organs. There are two otolith organs, the utricle and saccule, housing the sensory epithelium macula to detect linear acceleration (Fig. 1a, ut, sac), and three ampullae housing the crista to detect angular acceleration (Fig. 1a, ac, lc, pc). Each sensory organ exhibits regional specializations such as the striola in maculae and the central zone in cristae[1,2]. Specific functions attributed to these regional differences are just beginning to be realized[3,4]. For example, it has been demonstrated that the striola is required for detecting transient linear accelerations based on vestibular evoked potential (VsEP) measurements in mice, yet the lack of striolar and central zones do not appear to affect a mouse's ability to swim[5].

Another specialization in the maculae is the line of polarity reversal (LPR), across which the stereociliary bundles (hair bundles) of sensory hair cells (HCs) are oriented in opposite directions (Fig. 1b)[6,7].

[1]Section on Sensory Cell Regeneration and Development, Laboratory of Molecular Biology, National Institute on Deafness and Other Communication Disorders, National Institutes of Health, Bethesda, MD 20892, USA. [2]Mouse Auditory Testing Core Facility, National Institute on Deafness and Other Communication Disorders, National Institutes of Health, Bethesda, MD 20892, USA. [3]Department of Biomedical Engineering, Johns Hopkins University School of Medicine, Baltimore, MD 21205, USA. [4]Bioinformatics and Biostatistics Collaboration Core, National Institute on Deafness and Other Communication Disorders, National Institutes of Health, Bethesda, MD 20892, USA. [5]Department of Biology & Department of Otolaryngology, University of Iowa, Iowa City, IA 52242, USA. [6]Department of Special Education and Communication Disorders, 301 Barkley Memorial Center, University of Nebraska-Lincoln, Lincoln, NE 68583, USA. [7]Present address: Sensory & Motor Systems Research Group, Korea Brain Research Institute (KBRI), 61 Cheomdan-ro, Dong-gu, Daegu 41062, Republic of Korea. [8]Present address: Otolaryngology/Head and Neck Surgery, Kyoto University Hospital, 54 Shogoin-kawahara-cho, Sakyo-ku, Kyoto City, Kyoto 606-8507, Japan. [9]Present address: ENT Institute and Otorhinolaryngology Department of Eye & ENT Hospital, Fudan University, Shanghai 200031, China. ✉e-mail: wud@nidcd.nih.gov

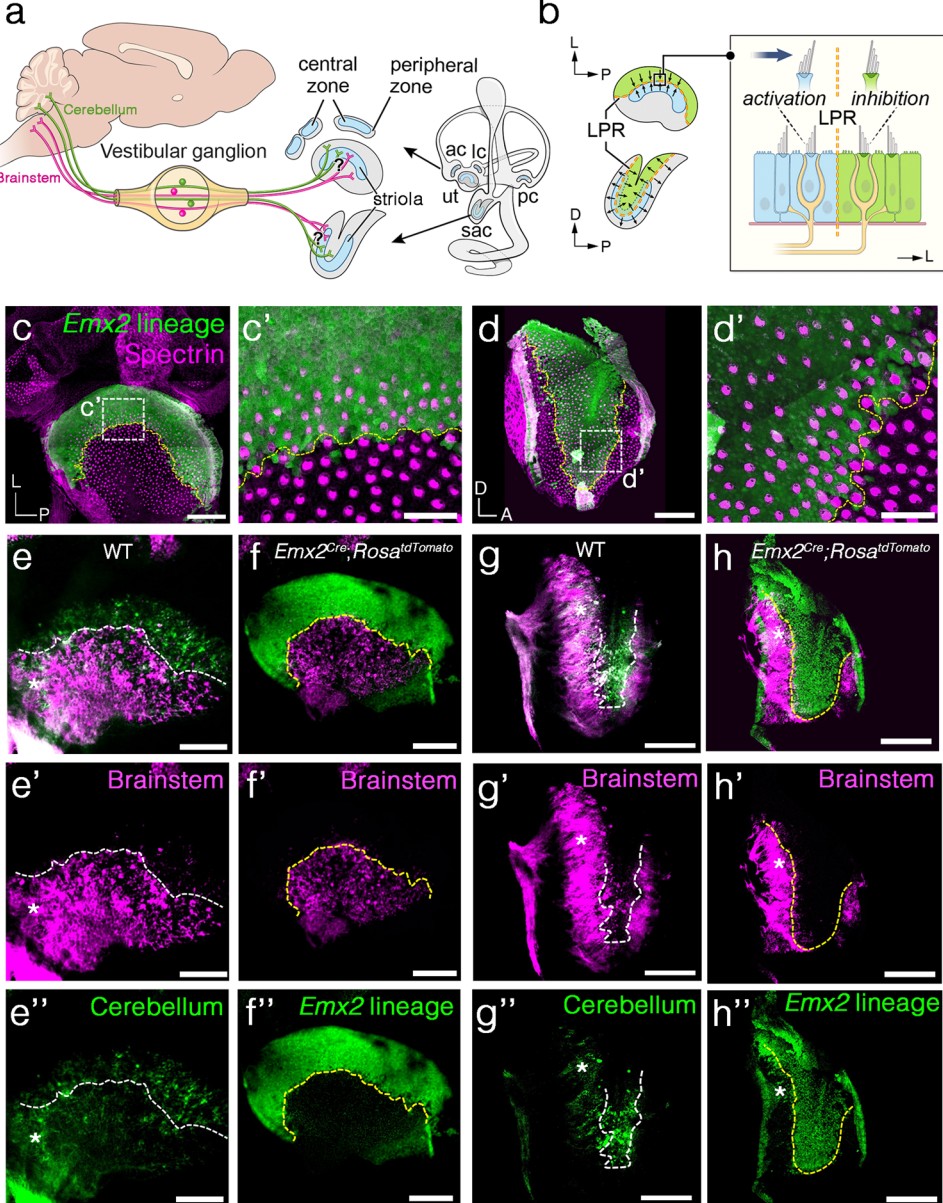

**Fig. 1 | The afferent neuronal innervation pattern is segregated across the LPR in the mouse maculae. a** Schematic of the segregated neuronal innervation pattern in maculae but the location of the segregation border is unclear. **b** Schematic illustrating the border of Emx2 expression domain (green) demarcates the LPR in maculae. A directional stimulus (arrow in inset) across the LPR simultaneously activates and inhibits two different groups of HCs. **c, d** *Emx2* lineage domain (green) in the utricle (**c, c′**, *n* = 2) and saccule (**d, d′**, *n* = 2) of *Emx2^cre/+^;Rosa^tdTomato^* inner ears, which encompasses HCs with opposite hair bundle orientation from the rest of the macula based on anti-β spectrin staining (magenta). The location of the tallest rod of the bundle, the kinocilium, is devoid of anti-β spectrin staining[54]. The border of the lineage domain demarcates the LPR (yellow dotted line). **e–h″** Neuronal tracing using NeuroVue® Red (magenta)- and NeuroVue® Maroon (green)-soaked filter papers inserted into the brainstem and cerebellum of hemi-sectioned heads of

wildtype (**e–e″**, **g–g″**, WT) and *Emx2^cre/+^; Rosa^tdTomato^* (**f–f″**, **h–h″**) embryos at E16.5, respectively. Neurons projected to the brainstem (**e′**, **g′**, magenta) and cerebellum (**e″**, **g″**, green) are segregated in the utricle (**e**, *n* = 5) and saccule (**g**, *n* = 3). White dotted line roughly separates most of the green and magenta dye labels. Limited dye labeling from the brainstem (**f′**, **h′**, magenta) is evident in the *Emx2* lineage domain (**f″**, **h″**, green) of the lateral utricle and inner saccule (**f**, *n* = 2; **h**, *n* = 2). White asterisk in **e–e″** and **g–g″** represents regions that show mixed labeling of both dyes that has been described previously[16]. White asterisk in **h–h″** represents a dorsal–posterior region of the saccule and where *Emx2* lineage domain expands beyond the LPR[13], where the two dye labels overlap (**g–g″**, white asterisk). ac anterior crista, lc lateral crista, pc posterior crista, sac saccule, ut utricle. Orientation: D dorsal, L lateral, P posterior. Scale bar = 100 μm in **c–h**; 25 μm in **c′, d′**.

This LPR of the maculae is conserved from fish to human[6,8–14] although little is known about its function. Given hair bundle orientation of HC confers the cell's directional sensitivity[10,15], the LPR presumably generates additional directional sensitivity to linear acceleration. Based on the position of the LPR within the maculae, any directional stimulus to a macula is likely to generate an activation and inhibitory signal simultaneously (Fig. 1b). The integration of these signals for vestibular functions is not known.

Notably, neuronal fiber tracing studies using lipophilic dyes have demonstrated that the vestibular neurons that innervate each macula are regionally segregated with different central projections to either the uvula and nodulus of the cerebellum or vestibular nuclei in the brainstem (Fig. 1a)[16]. This segregation at the macula begins at embryonic day (E) 14.5, established by E15.5, and persisted when examined at postnatal day (P) 21[14]. Although there are obvious functional implications, it is not known whether the peripheral innervation

pattern is segregated across the LPR since dye-traced samples are not compatible with immunostaining pre- or post-dye labeling.

Previously, we have shown that the transcription factor Emx2 mediates the LPR formation. The restricted expression of Emx2 to the lateral region of the utricle and the inner region of the saccule causes hair bundle reversal within its expression domain and thus generates the LPR (Fig.1b)[13]. This function of Emx2 is conserved as it mediates hair bundle reversal in HCs of neuromasts in the zebrafish lateral line system[13,17,18]. Notably, Emx2 expression in HCs of the neuromast also regulates the selection of the innervating neurons[19]. Whether Emx2 is required for neurons to innervate macular organs in a regionally segregated manner in the mammalian vestibular system is not known.

In this study, we asked whether the neuronal innervation pattern in the maculae is indeed segregated across the LPR. Our results show that Emx2 mediates bidirectional sensitivity on two levels, hair bundle orientation in HCs and neuronal selection by HCs and supporting cells. Furthermore, we investigated the functional significance of the bidirectional sensitivity by generating two mutants: *Emx2* conditional knockout mice (cKO), which lacks the LPR and *Emx2*<sup>cre</sup>; *Tmie*<sup>F/−</sup> (*Tmie* cKO), in which the LPR is present but mechanotransduction of all HCs within the *Emx2* expression domain failed. Behavioral results from both *Emx2* cKO and *Tmie* cKO indicate that the bidirectional sensitivity of maculae is required for challenging vestibular functions such as traversing efficiently on a narrow balance beam and swimming comfortably.

## Results

### Afferent innervation in normal maculae is segregated across the LPR

Innervations of vestibular afferent neurons in the maculae are segregated based on their central projections (Fig. 1e–e", g–g")[16]. In normal maculae, NeuroVue® Maroon dye from the cerebellum labels the lateral utricle (Fig. 1e, e", green color) and inner saccule (Fig. 1g, g", green color), whereas NeuroVue® Red from the brainstem labels the medial utricle (Fig. 1e, e', magenta color) and outer saccule (Fig. 1g, g', magenta color). Emx2 reverses hair bundle orientation in the maculae and the border of the *Emx2* lineage domain demarcates the LPR (Fig. 1c, c', d, d', yellow dotted line)[13]. Both *Emx2* expression or its lineage domain (Fig. 1f, f", h, h", converted to green color)[13] appears to coincide with dye tracing from the cerebellum in the lateral utricle (Fig. 1e, e", green color) and inner saccule (Fig. 1g, g", green color). To determine whether the afferents segregate at the LPR, we conducted neuronal tracing from the brainstem in *Emx2*<sup>cre</sup>; *Rosa*<sup>tdTomato</sup> embryos at E16.5, in which the *Emx2* lineage domain is tdTomato-positive. The border of the tdTomato reporter domain demarcates the LPR without the need of immunostaining. We found that the *Emx2* lineage domain (Fig. 1f, f", h, h", green color) was complementary to dye labeling from the brainstem in the medial utricle (Fig. 1f, f', magenta color) and outer saccule (Fig. 1h, h', magenta color). These results indicate that normal afferent neurons are segregated across the LPR in the maculae and neurons that innervate HCs within the *Emx2* domain project to the cerebellum, whereas those that innervate HCs outside of the *Emx2* domain project to the brainstem.

### Afferents projecting to the cerebellum fail to reach *Emx2* KO maculae

The segregation of neuronal innervation pattern across the LPR in maculae prompted us to ask whether Emx2 is required for neuronal segregation, and we compared neuronal dye-tracing between control and *Emx2* KO maculae (Fig. 2 and Supplementary Fig. 1). In control ears, dyes reached the sensory epithelium of the maculae and cristae after 7 days of incubation. NeuroVue® Maroon delivered to the cerebellum labeled the cristae and Emx2-positive−lateral utricle and inner saccule (Fig. 2a–b" and Supplementary Fig. 1a, green) and NeuroVue® Red delivered to the brainstem labeled the cristae, medial utricle, and outer

saccule (Fig. 2a–b" and Supplementary Fig. 1a, magenta). Some of the nerve fibers in the sensory epithelium of the utricle appeared vacuolated and they were likely to be the labeled-calyces surrounding cell bodies of Type I HCs[1,20] (Fig. 2b', b", yellow arrowhead and insets). However, NeuroVue® Maroon dye from the cerebellum (green), which normally reaches the lateral utricle and inner saccule was much reduced in *Emx2* KO maculae (Fig. 2c–d" and Supplementary Fig. 1b), even though this dye reached the anterior crista (does not normally express Emx2[13]) in a similar manner to controls (Fig. 2a, c, white arrowheads). By contrast, in *Emx2* KO utricles, dye-labeling from the brainstem was expanded into the lateral region of which is normally innervated by neurons that project to the cerebellum. These ectopic brainstem fibers showed similar vacuolated pattern in the lateral region (Fig. 2d', yellow arrowhead and inset) as controls (Fig. 2b', b"), suggesting that the ectopic brainstem fibers reached the sensory epithelium in a comparable manner to the endogenous cerebellum-projected fibers (Fig. 2e, f).

Although it is difficult to determine whether the brainstem afferents reached the lateral edge of the sensory epithelium of *Emx2* KO utricles in the dye-traced specimens, Tuj1 staining of similar utricular sections showed comparable presence of nerve fibers in the lateral region of control and mutant utricles (Supplementary Fig. 2). These results suggest that in *Emx2* KO maculae, neurons that project to the brainstem indeed have their peripheral branches reaching the lateral edge of the utricle replacing the nerve fibers of neurons that normally project to the cerebellum centrally. In support of the results observed in utricles, dye traced from the brainstem also expanded into the innermost region of the *Emx2* KO saccule (Supplementary Fig. 1, white asterisk). Together, these results suggest that normal *Emx2* expression domain in the lateral utricle and inner saccule promotes peripheral targeting of vestibular neurons, which project their central fiber to the cerebellum.

### Induction of apoptosis in vestibular ganglion of *Emx2* KO

Next, we addressed whether the lack of peripheral targeting of vestibular neurons in the *Emx2* KO affected neuronal survival. Since the identity of the neurons that normally innervate HCs within the Emx2 domain is not known, we assessed overall activated caspase-3 positive cells in the vestibular ganglion between E16.5 and E18.5 (Supplementary Fig. 3). We found significantly more active caspase-3-positive cells in the vestibular ganglion of *Emx2* KO in each of these ages compared to controls (Supplementary Fig. 3), suggesting that some of the afferent neurons that normally project to the cerebellum were undergoing apoptosis most likely because of their failure to reach their peripheral targets.

### Mild disruption of neuronal segregation pattern in *Emx2* cKO utricles

Since *Emx2* expression spans from luminal to basal side of the sensory epithelium[13,21], we addressed whether *Emx2* expression in HCs alone is required for establishing both hair bundle orientation and neuronal selectivity by using *Gfi1*<sup>cre22</sup>, a HC-specific *cre* strain and a floxed allele (F) of *Emx2* (Fig. 3). Compared to controls (Fig. 3a, a'), hair bundles in the lateral region of *Gfi1*<sup>cre</sup>; *Emx2*<sup>F/−</sup> (*Emx2* cKO) utricles where *Emx2* is normally expressed were reversed and there was no LPR (Fig. 3b, b', f). However, the neuronal innervation pattern in *Emx2* cKO utricles remained largely segregated except for the border separating the two dyes was not as well demarcated as in controls (Fig. 3c, d).

To quantify the extent of this neuronal disruption in the utricle, the intensities of dyes from the cerebellum (cb) over total signals from both cerebellum and brainstem (cb + bs, see "Methods") were compared using *z*-stacks of confocal images from three representative regions along the anterior−posterior axis of the control and mutant utricles (Fig. 3e, dotted lines in schematic). Each of the selected sensory regions was further divided into three equal regions with the

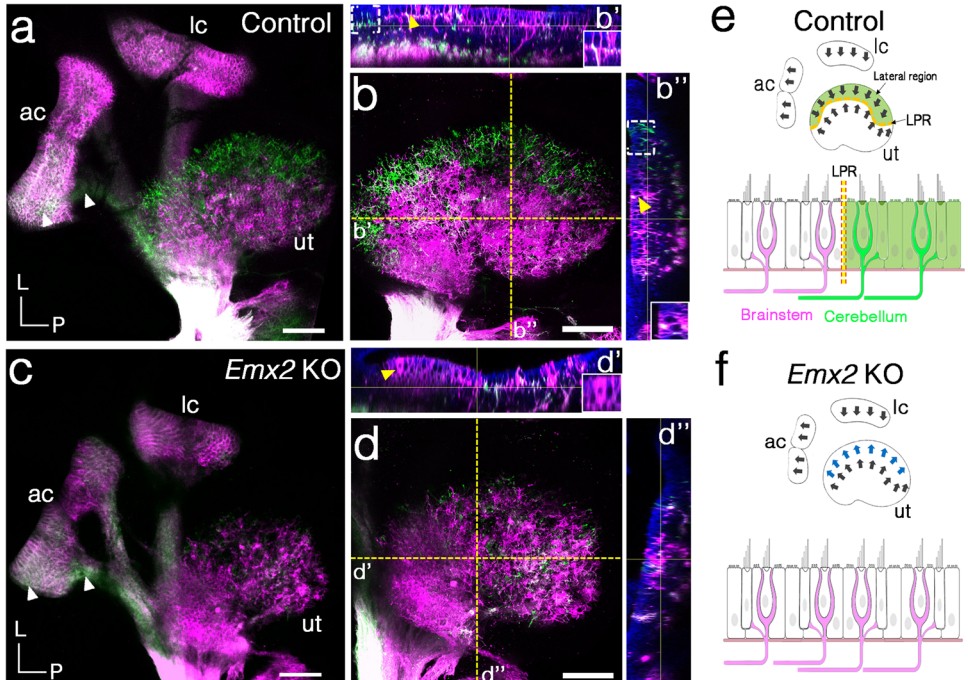

**Fig. 2 | Failure of cerebellar afferent neurons to reach *Emx2* KO maculae. a–b"**, **c–d"** Neuronal tracing with NeuroVue® Red (magenta)- and NeuroVue® Maroon (green)-soaked filter papers inserted into the respective brainstem and cerebellum of hemi-sectioned heads of control (**a–b"**) and *Emx2* KO (**c–d"**) at E16.5. **b**, **d** are higher magnifications of the maximum-intensity projection images of **a**, **c**. **b'**, **b"**, **d'**, **d"** are z-stack images of areas indicated with yellow dotted lines in **b**, **d**, respectively. In control, dye traced from the cerebellum (green) is segregated to the lateral region of utricle (**b'**, **b"**, white dotted bracket), whereas tracing from the brainstem is largely restricted to the medial utricle (**a**, n = 5). Dyes from both cerebellum and brainstem reached the sensory epithelium with some appeared vacuolated, which are likely due to surrounding bodies of HCs (**b'**, **b"**, yellow arrowheads, see insets). In *Emx2* KO utricles (**c**, n = 7), dye traced from the cerebellum (green) is much reduced and failed to reach the lateral utricle, whereas this region is filled with dye traced from the brainstem (magenta), some of which show a vacuolated appearance (**d'**, yellow arrowhead, inset), similar to controls (**b'**, **b"**). White arrowheads in **a**, **c** point to dye from the cerebellum reaching the anterior crista in both control and *Emx2* KO ears. **e**, **f** Schematic summary of hair bundle orientation and afferent neuronal innervation pattern in control and *Emx2* KO utricles. Anterior crista (ac), lateral crista (lc), and utricle (ut). Scale bar = 100 μm in **a–d**.

lateral one-third defined as the lateral region (area #1, 4, 7) and the middle one-third as the intermediate_medial region (intermed_med, areas #2, 5, and 8) and medial one-third as the medial region (areas #3, 6, and 9). Since the exact position of the LPR could not be determined easily in these dye-traced specimens, we reasoned that there should be a significant decrease in the relative cb signal between the lateral and its corresponding intermed_medial region if this assignment of lateral and medial regions across the LPR was correct.

To investigate this hypothesis, we performed multiple linear regression analysis to elucidate the relationship between cb signals and explanatory variables region and genotype (see "Methods"). Four mixed control utricles were selected for this analysis and the variability of their dye labeling are illustrated in Supplementary Fig. 4. The linear regression model explained a significant amount of variance in the value of the cb signal ($F(5,84) = 53.08$, $R^2 = 0.76$, Adjusted $R^2 = 0.75$, $p < 2.2E-16$) and the analysis showed that there was a consistent decrease in the cb signals between the lateral regions and the adjacent intermed_medial ($\beta = -0.59$, 95% CI [−0.73, −0.46], s.e. = 0.068, $p = 1.5E-13$) or the medial regions ($\beta = -0.67$, 95% CI [−0.80, −0.53], s.e. = 0.068, $p = 1.2E-15$) in both the control and *Emx2* cKO utricles (Fig. 3e and Supplementary Note 1). These results indicated that our assignment of lateral and medial regions was reasonable and the neuronal segregations between cKO and control utricles were similar. The cb signal appeared to increase in the *Emx2* cKO utricles compared to controls, but it was only significant in the intermed_medial region (significant interaction term, intermed_medial: $Gfi1^{Cre}$; $Emx2^{F/-}$) (Fig. 3e and Supplementary Note 1), $\beta = 0.18$, 95% CI [0.0076, 0.35], s.e = 0.087, $p = 0.041$; Kruskal–Wallis $H(1) = 9.82$, $p = 1.7E-03$). These results confirmed the imaging results that the neuronal segregation in *Emx2* cKO was largely similar to

controls except with disruption occurring at the border where the two types of neurons normally segregate (Fig. 3f).

Furthermore, these results suggest that while Emx2 expression in the HCs regulates proper neuronal targeting in the maculae, the source of Emx2 in the supporting cells is likely to be required for this function as well. In that regard, we tried two *cre* lines for supporting cells, $Gfap^{cre}$ [23] and $Plp^{creER}$ [24,25], but neither of these lines conferred reasonable cre reporter expression in supporting cells of the lateral utricle. Short of an effective *cre* line targeting supporting cells, the role of supporting cells in neuronal targeting cannot be addressed directly. Nevertheless, the mild disruption of neuronal phenotype but complete penetrance of the hair bundle phenotype in *Emx2* cKO suggests that the regulation of neuronal innervation and hair bundle orientation by Emx2 may be independent of each other.

**Ectopic *Emx2* disrupts neuronal segregation in the utricle**

Since endogenous *Emx2* regulates the neuronal innervation in the maculae, we asked whether ectopic expression of *Emx2* across the maculae affects the innervation as well. Compared to controls (Fig. 4a, a'), ectopic expression of *Emx2* in the sensory epithelium of $Sox2^{creER}$; $Rosa^{Emx2}$ embryos by administering tamoxifen at E13.5 reversed hair bundle orientation in the medial utricle without affecting hair bundle orientation in the endogenous Emx2 domain (Fig. 4b, b', white arrows abnormal orientation)[13]. However, this effect was much diminished when tamoxifen was administered at E15.5 (Fig. 4c, c', white arrows abnormal orientation)[13]. By contrast, the neuronal innervation pattern appeared equally affected when *Emx2* was activated with tamoxifen administration on either E13.5 or E15.5 (Fig. 4d–f).

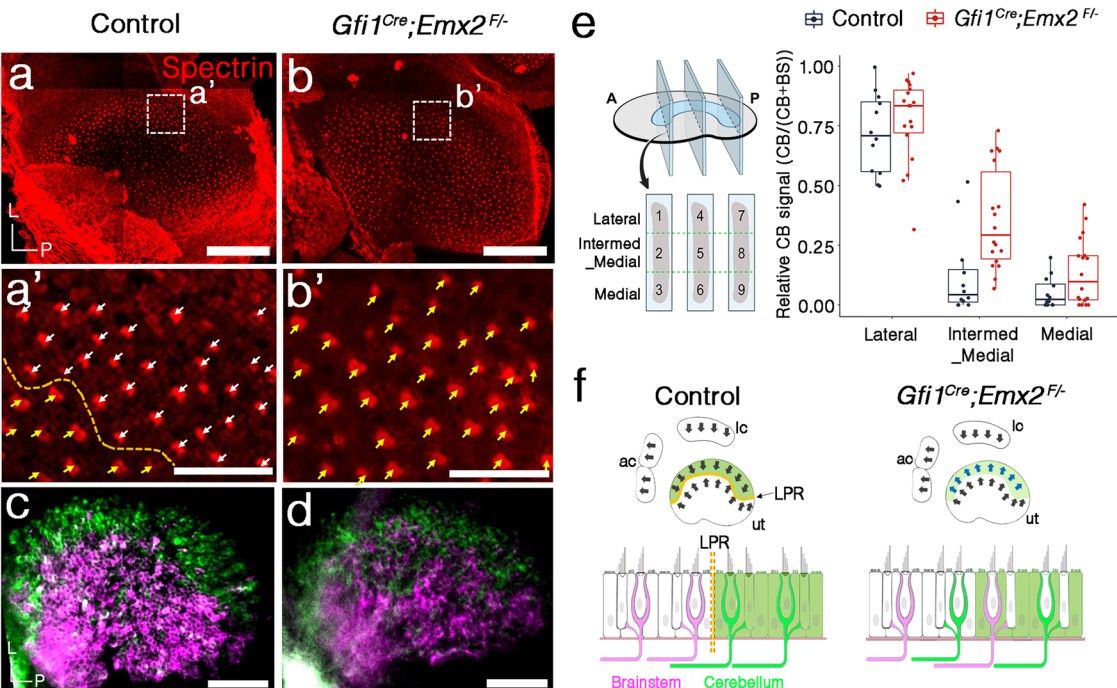

**Fig. 3 | Minor disruption of neuronal innervation pattern in HC-specific *Emx2* KO. a–b'** Hair bundle orientation across the LPR (yellow dotted line) of control (**a**, **a'**) is missing in the *Emx2* cKO. Hair bundle orientation in the lateral region of *Emx2* cKO utricles is similar to its medial region (**b**, **b'**, yellow arrows, *n* = 4) and is reversed from the lateral region of controls (**a**, **a'**, white arrows, *n* = 4). **c**, **d** Neuronal dye tracing from the cerebellum (green) and brainstem (magenta) are not well segregated in the *Emx2* cKO utricles (**d**, *n* = 9) as in controls (**c**, *n* = 5). **e** Boxplot showing the quantification of the cb signal from the cerebellum over total signals from cerebellum and brainstem using *z*-stacks of confocal images in nine selected regions according to the "Methods" section. In control utricles, the relative cb signal is significantly higher in the lateral utricle (areas #1, 4, and 7) than the intermediate_medial (areas #2, 5, and 8) and the medial region (areas #3, 6, and 9) (Dunn's pairwise tests, *z* = 3.83, *p* = 2.6E−04; *z* = 4.45, *p* = 2.5E−05). Similar decrease

in relative cb signal was observed from lateral to intermediate-medial and the medial regions of *Emx2* cKO utricles (Dunn's pairwise tests, *z* = 3.57, *p* = 7.1E−04; *z* = 5.90, *p* = 1.1E−08). However, the relative cb signal in the intermed_medial region is higher in *Emx2* cKO utricles, compared to control (Kruskal−Wallis *H*(1) = 9.82, *p* = 1.7E−03). Controls, *n* = 4; *Gfi1^{Cre}; Emx2^{F/−}*, *n* = 6. The two-sided Kruskal−Wallis rank-sum test and two-sided post hoc Dunn's pairwise comparison tests were applied. In boxplots, boxes represent the interquartile range (IQR), and the thick lines inside show the median. Whiskers denote the lowest and highest values within 1.5 times the IQR. **f** Schematic summary of hair bundle orientation and afferent neuronal innervation pattern in control and *Gfi1^{Cre}Emx2^{F/−}* (*Emx2* cKO) utricles. *Gfi1^{Cre}; Emx2^{F/−}* utricles show uni-directional HCs and mild disruption of the neuronal innervation pattern, compared to controls. Scale bar = 100 μm in **a**–**d**; 25 μm in **a'**−**b'**. Source data are provided as a Source data file.

To be able to compare the extent of the neuronal phenotypes among the various mutant strains, we compared the relative cb signal of the *Sox2^{creER}; Rosa^{Emx2}* utricles treated with tamoxifen at E13.5 or E15.5 against the same selected controls used for the *Emx2* cKO (Fig. 3e and Supplementary Fig. 4, see "Methods"). If ectopic *Emx2*'s effect on neuronal innervation is similar to its effect on hair bundle orientation[13], one would expect the relative cb signal to increase in the medial region but remain relatively normal in the lateral region. In contrast, compared to controls, our results showed that the cb signal was affected in the entire utricle: a significant decrease of cb signal in the lateral region and increase in the intermed_medial region of mutants regardless of the age of tamoxifen injection (Fig. 4g and Supplementary Note 2). As a result, the normal decrease in cb signal from the lateral to intermed-medial regions observed in controls was not observed in *Sox2^{creER}; Rosa^{Emx2}* utricles (Fig. 4g and Supplementary Note 2). These results indicate that ectopic *Emx2* has a longer time window on changing afferent neuronal innervation than changing hair bundle orientation (Fig. 4h), and further suggests that these two effects of Emx2 are independent from each other.

## Emx2 in both HCs and supporting cells regulate afferent innervation

The lack of a good *cre* strain for the supporting cells within the Emx2 domains negates the use of a conditional knockout approach to assess the role of supporting cells in regulating neuronal innervation. Thus, we asked whether ectopic expression of *Emx2* selectively in HCs or

supporting cells could disrupt neuronal innervation by using a HC-specific and a supporting cell-specific cre strain, *Gfi1^{Cre}* and *Plp^{CreER}*, respectively. While ectopic expression of *Emx2* in HCs (Fig. 5b, b')[13] but not supporting cells reversed hair bundle orientation (Fig. 5c, c'), both mutant strains showed disrupted neuronal innervation pattern (Fig. 5e, f), compared to controls (Fig. 5a, a', d). The normal decrease in cb signal between lateral and intermed_medial region was only significant in *Plp^{CreER}; Rosa^{Emx2}* and not *Gfi1^{Cre}; Rosa^{Emx2}* utricles treated with tamoxifen at E13.5, suggesting that the neuronal disruption by ectopic *Emx2* is more severe in HCs than supporting cells. Compared to controls, both *Gfi1^{Cre}; Rosa^{Emx2}* and *Plp^{CreER}; Rosa^{Emx2}* utricles did not show a significant difference in cb signals in the lateral regions but an increase in cb signals in the intermed_medial and medial regions (Fig. 5g and Supplementary Note 3), indicating that the afferents were primarily affected in the medial regions. While the difference in hair bundle polarity phenotype observed between the two ectopic *Emx2* mouse mutants (Fig. 5b, b')[13] supported the notion that the intended cell type was being targeted by the respective *Gfi^{cre}* and *Plp^{creER}* strain, we cannot preclude the possibility that other ectopic source(s) of *Emx2* such as the brain and glial cells surrounding the vestibular nerve could confound the observed neuronal phenotype[22,25]. Nevertheless, the ability of ectopic *Emx2* in either HCs or supporting cells in disrupting afferent neuronal innervation further distinguishes this role of Emx2 from its role in altering hair bundle orientation, which is required in HCs autonomously (Fig. 3b, b').

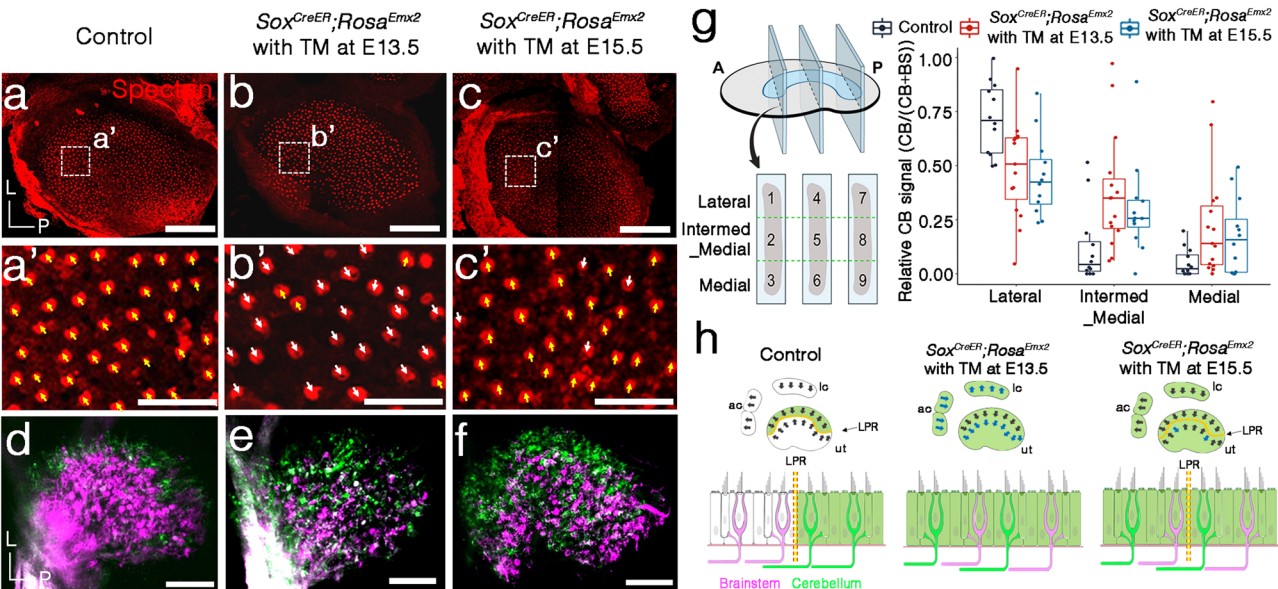

**Fig. 4 | Ectopic *Emx2* expression disrupts afferent neuronal segregation pattern independent of hair bundle orientation in the utricle. a, a'** Anti-β spectrin antibody (red) staining of a control utricle showing hair bundles in the medial region pointing towards the lateral (**a'**, yellow arrows, n = 4). Compared to the control, *Sox2^CreER; Rosa^Emx2* utricle administered with tamoxifen (TM) at E13.5 (**b**) shows mostly reversed hair bundle orientation in the medial region (**b'**, white arrows, n = 4) but TM administration at E15.5 (**c**) resulted in largely normal hair bundle orientation (**c'**, yellow arrows, n = 4) with a few HCs showing opposite orientation (white arrows). **d**–**f** Segregated afferent innervation of control (**d**, n = 5) is disrupted in *Sox2^CreER; Rosa^Emx2* utricles administered with TM at E13.5 (**e**, n = 10) and E15.5 (**f**, n = 7). **g** The relative cb signal was significantly lower in the lateral region (Dunn's pairwise tests, control vs. E13.5, n = 5, z = 2.59, p = 0.015; control vs. E15.5, n = 4, z = 3.04, p = 0.007) but higher in the intermed–medial region of

*Sox2^CreER; Rosa^Emx2* utricles treated with tamoxifen at E13.5 or E15.5 when compared to controls (n = 4) (Dunn's pairwise tests, control vs. E13.5, z = −3.0, p = 0.008; control vs. E15.5, z = −2.23, p = 0.039). Thus, the decrease in cb signals between the lateral and intermed_medial regions were not significant in *Sox2^CreER; Rosa^Emx2* utricles administered with tamoxifen at either E13.5 or E15.5 (Dunn's pairwise tests, E13.5, z = 1.21, p = 0.23; E15.5, z = 1.90, p = 0.12). The two-sided Dunn's pairwise comparison tests were applied. In boxplots, boxes represent the interquartile range (IQR), and the thick lines inside show the median. Whiskers denote the lowest and highest values within 1.5 times the IQR. **h** Schematic summary of hair bundle orientation and innervation pattern in control and *Sox2^CreER; Rosa^Emx2* utricles and cristae administered with tamoxifen at E13.5 or E15.5. Scale bar = 100 μm in **a**–**c**, **d**–**f**; 25 μm in **a'**–**c'**. Source data are provided as a Source data file.

## Normal neuronal innervation in *Tmie* cKO

Hair bundle orientation is unidirectional and LPR is absent in maculae of either gain or loss of function of *Emx2* within HCs. The gain-of-function mutant, *Gfi1^Cre; Rosa^Emx2*, shows severe vestibular deficits presumably due to hair bundle orientation defect in both maculae and cristae[13] as well as innervation defects (Fig. 5e), and these mice die at early postnatal ages. In contrast, the fully penetrant hair bundle defects but mild disruption of neuronal innervation pattern in the viable *Emx2* cKO makes it a good model to address the functional significance of the LPR. However, in the *Emx2* cKO maculae, the hair bundle reversal causes a loss of sensitivity to stimuli from some directions but a concomitant gain of HCs responding to the opposite directions of stimuli. This gain in HC responses, however, is being relayed to the cerebellum rather than the brainstem, and the responses are opposite to what would be received by the cerebellum of controls from the same directional stimulus (see below). Thus, it may be difficult to discern whether the behavioral deficits observed in this mutant strain necessarily represents a loss of function. Therefore, we utilized *Tmie* to engineer another mouse without LPR function. *Tmie* encodes a protein that is required for mechanotransduction channel assembly and function in HCs[26,27], and mutations of this gene are associated with human deafness[28]. Thus, in *Tmie* cKO (*Emx2^cre; Tmie^F/−*), the LPR is present but HCs within the Emx2-positive domain are rendered non-functional leaving other HCs in the inner ear intact. As a control, we also engineered a sensory organ-wide knockout of *Tmie* using *Gfi1^cre*, which affected mechanotransduction channels in all HCs of the inner ear. Both the *Tmie* cKO and *Gfi1^Cre; Tmie^F/−* utricles showed a region-appropriate loss of *Tmie* transcripts in the lateral and the entire utricle, respectively (Fig. 6a–c'). FM 1-43 dye uptake, which quickly enters HCs via the mechanotransduction channels at the tips of short-row

stereocilia (Fig. 6d), was absent in the entire *Gfi1^Cre; Tmie^F/−* utricles (Fig. 6e) but was only absent from the Emx2-positive, lateral region of the *Tmie* cKO utricles (Fig. 6f, yellow dotted line). Nevertheless, the neuronal innervation pattern appeared normal in both mutant strains compared to controls at E16.5 (Fig. 6g–I and Supplementary Fig. 5) when the innervation pattern was initially established, as well as later at postnatal day 0 (Fig. 6j–l, P0, Supplementary Fig. 5). These results suggest that HC activity is not required for establishing the afferent neuronal innervation pattern in the maculae. Furthermore, *Tmie* cKO, which show selective loss of HC functions within the Emx2-positive domains of the maculae but without detectable defects in the neuronal segregation, presents a lack of bidirectional sensitivity model without a gain of sensitivity to the remaining directional stimuli (see below).

## Bidirectional sensitivity facilitates specific vestibular functions

While the *Gfi1^Cre; Tmie^F/−* mice exhibited severe vestibular deficits such as head bobbing, head tilt and circling phenotypes similar to the *Tmie* KO[29], both the *Emx2* cKO and *Tmie* cKO mutants appeared normal and behaved indistinguishably from their littermates in the mouse cage. To investigate the behavioral consequence of losing bidirectional sensitivity in the maculae, we subjected the *Emx2* cKO and *Tmie* cKO mutants to the open-field test and several challenging vestibular activities such as staying on a rotating rod, traversing a balance beam, and forced swimming. Both cKO mutants did not show any hyperactivity in the open-field test under dark conditions (Supplementary Fig. 6). With increasing acceleration of a rotating rod, the controls and the two mutant strains improved their ability to stay on the rotating rod over the 3-day trial. While there was a tendency for *Tmie* cKO to fall off the rotating rod sooner on the first day of trial, they improved on subsequent trial days and there was no significant difference in the

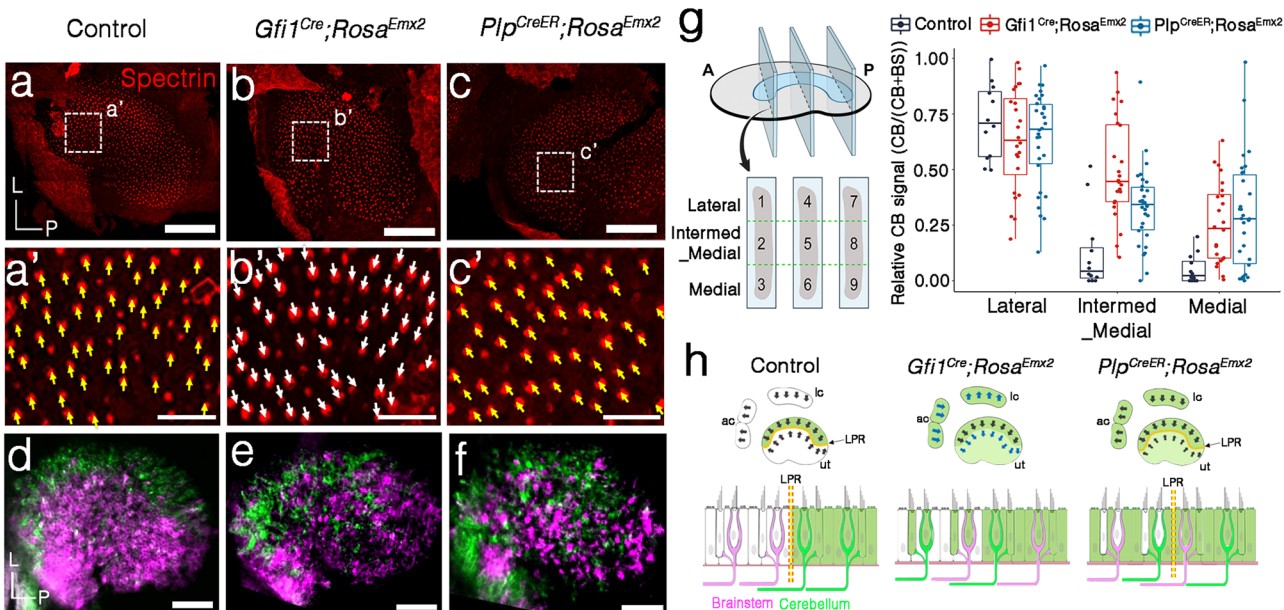

**Fig. 5 | Afferent neuronal segregation pattern in the utricle is affected by ectopic *Emx2* expression in either HCs or supporting cells. a–c′** Anti-β spectrin staining showing hair bundle orientation in the medial utricle is towards the lateral, which is reversed in *Gfi1^Cre^; Rosa^Emx2^* utricles (**b′**, white arrows, *n* = 3) but normal in *Plp^CreER^; Rosa^Emx2^* utricles (**c′**, yellow arrows, *n* = 5). **d–f** Neuronal dye tracing from the cerebellum (green) and brainstem (magenta) in control (**d**, *n* = 5), *Gfi1^Cre^; Rosa^Emx2^* (**e**, *n* = 10) and *Plp^CreER^; Rosa^Emx2^* utricles treated with tamoxifen at E13.5 (**f**, *n* = 13). Neuronal dye tracing is not well segregated in *Gfi1^Cre^; Rosa^Emx2^* (**e**) and *Plp^CreER^; Rosa^Emx2^* utricles (**f**) as in controls (**d**). **g** The reduction in relative cb signal from lateral to intermed_medial region was not significant for *Gfi1^Cre^; Rosa^Emx2^* utricles (*n* = 8) but significant for *Plp^CreER^; Rosa^Emx2^* utricles (*n* = 10) (Dunn's pairwise tests, *Gfi1^Cre^; Rosa^Emx2^*, *z* = 1.70, *p* = 0.09; *Plp^CreER^; Rosa^Emx2^*, *z* = 4.29, *p* = 3.5E−05). Compared

to controls (*n* = 4), there was no significant difference in the relative cb signal in the lateral region but a significant increase in the intermed_medial and medial regions of *Gfi1^Cre^; Rosa^Emx2^* and *Plp^CreER^; Rosa^Emx2^* utricles (control vs. *Gfi1^Cre^; Rosa^Emx2^*, intermed_medial, *z* = -4.45, *p* = 2.6E−05; medial, *z* = -3.37, *p* = 0.0011; control vs. *Plp^CreER^; Rosa^Emx2^*, intermed_medial, *z* = -2.65, *p* = 0.012; medial, *z* = -3.53, *p* = 0.0013). The two-sided Dunn's pairwise comparison tests were applied. In boxplots, boxes represent the interquartile range (IQR), and the thick lines inside show the median. Whiskers denote the lowest and highest values within 1.5 times the IQR.
**h** Schematic of hair bundle orientation and afferent neuronal innervation pattern in control, *Gfi1^Cre^; Rosa^Emx2^* and *Plp^CreER^; Rosa^Emx2^* utricles. Scale bar = 100 μm in **a–c**, **d–f**; 25 μm in **a′–c′**. Source data are provided as a Source data file.

---

ability of the two mutant strains to stay on the rotarod among the 3 days of trials, compared to the controls (Fig. 7a).

For testing the ability of mice to traverse a balance beam, a balance beam located 80 cm above ground was used. All mice tested had no trouble traversing a 12 mm-wide balance beam. On a 6-mm wide balance beam, the *Tmie* cKO showed a delay in traversing an 80 cm distance and one mutant failed to complete the task (*n* = 1/21), compared to controls (Fig. 7b). All but one of the *Emx2* cKO mutants completed the task of moving across the 6-mm balance beam (*n* = 1/24), compared to controls (Fig. 7b).

In a forced swim test, both mutant strains could swim and did not need to be rescued from the water. However, compared to controls, the mutants appeared to be frantic swimmers and they spent more time in trying to climb out of the water, which was noted independently by five observers blinded to the genotype. Thus, the percentage of time a mouse spent swimming vertically was quantified using the ForcedSwimScan software and both mutant strains showed a significant increase in the climbing time (swimming vertically) during the 1 min trial compared to controls (Fig. 7c). Taken together, these results indicate that *Emx2* cKO and *Tmie* cKO have subtle but specific vestibular deficits in water and traversing on a narrow beam. These deficits are attributed to defects in the otolith organs since Emx2 is not expressed in the three cristae[13], and the normal angular vestibular ocular reflex (aVOR) and optokinetc reflex (OKR) in these mutants support this hypothesis (Supplementary Figs. 7 and 8).

### Differential responses to directional jerk stimuli in *Emx2* cKO
While *Tmie* cKO showed delay in moving on the balance beam and both *Emx2* cKO and *Tmie* cKO struggle in water, it is difficult to ascribe these complex behavioral deficits specifically to loss of the LPR. Thus,

we took advantage of the vestibular evoked potential (VsEP) measurement, which represents summed far-field afferent responses to transient linear accelerations and can be elicited from two different directions of jerk stimuli from either nasal or occipital direction when the mouse is mounted in a supine position (Fig. 8a). During the positive direction of jerk stimulation (i.e., from occipital towards the nasal direction), both *Emx2* cKO and *Tmie* cKO showed lower amplitudes at maximal stimulus level (+6 dB re: 1 g/ms) than controls (Fig. 8c, d). Given both mutant strains showed reduced amplitudes in the positive direction of stimulation (from occipital towards the nasal direction), we predicted that under the negative (opposite) direction of stimulation, only *Emx2* cKO should show a higher amplitude but no change in amplitude for the *Tmie* cKO because only *Emx2* cKO has a gain of HC response to the opposite direction of stimuli (Fig. 8b). The similarity in amplitudes between controls and *Tmie* cKO in the negative direction of stimulus is consistent with our hypothesis. While *Emx2* cKO showed an increase in amplitudes in the negative direction of stimulus compared to controls, this was not a statistically significant increase (Fig. 8c, d). By averaging both the positive and negative VsEP responses, only *Tmie* cKO showed a lower amplitude compared to controls and not when controls and *Emx2* cKO were compared. We interpreted the decrease in the average amplitude of *Tmie* cKO but no change in the average amplitude of the *Emx2* cKO to mean that the slight increase in amplitudes in the negative stimulation direction of *Emx2* cKO, although not statistically significant, compensated for the reduced amplitudes in the positive stimulation direction resulting in no net change in amplitudes when the two directional stimulations were summed. Such a compensation was not observed in *Tmie* cKO, which showed a decreased average amplitude compared to controls. Consistently, the lower amplitudes observed in the two mutant strains in the positive direction

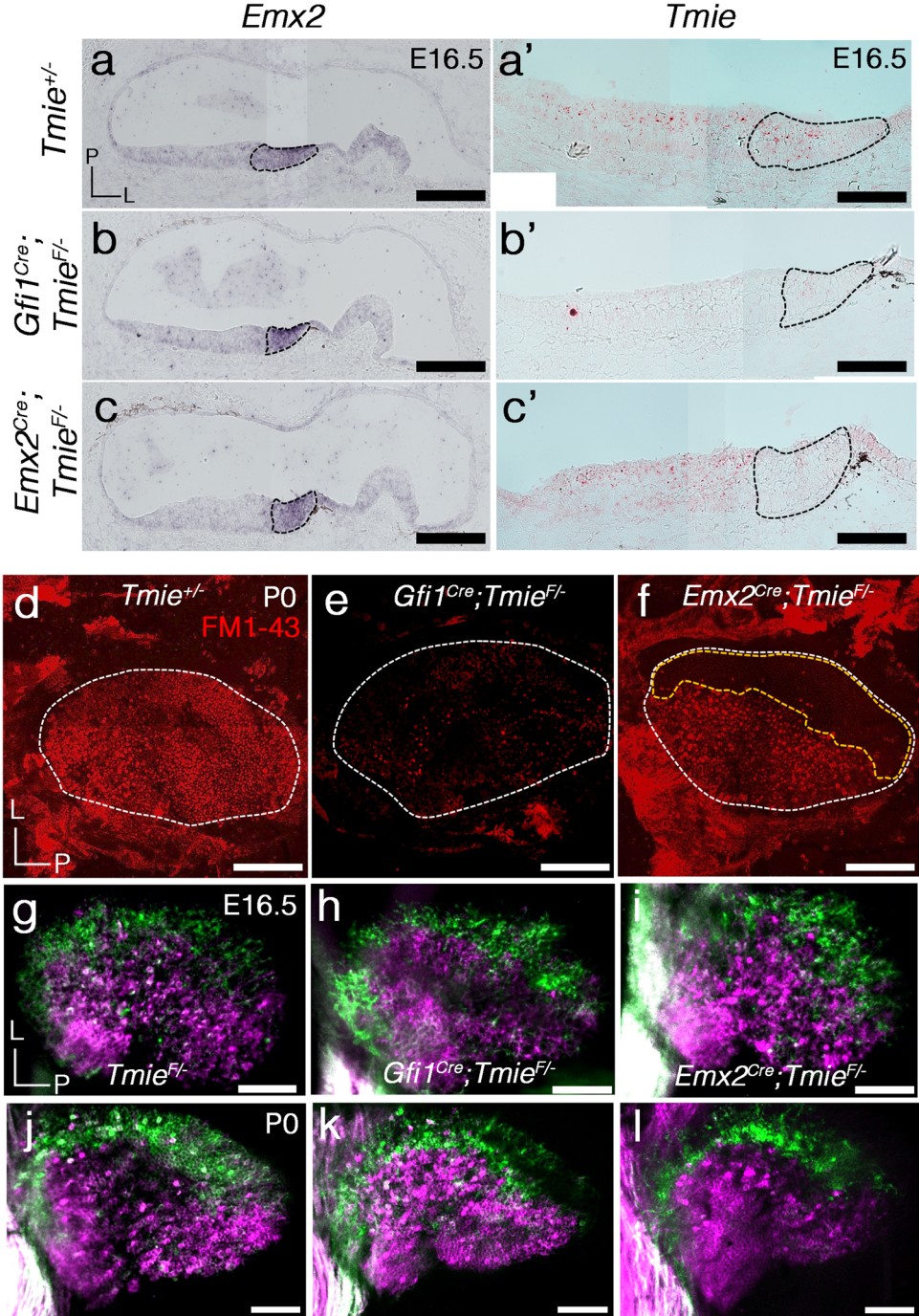

**Fig. 6 | Loss of mechanotransduction in HCs of *Tmie* cKO utricles does not affect its neuronal segregation pattern. a–c'** *Emx2* (**a–c**) and *Tmie* (**a'–c'**) transcripts in sections of *Tmie*⁺/⁻ (**a, a'**, *n* = 3), *Gfi1*ᶜʳᵉ; *Tmie*ᶠ/⁻ (**b, b'**, *n* = 2), and *Emx2*ᶜʳᵉ; *Tmie*ᶠ/⁻ (*Tmie* cKO, **c, c'**, *n* = 3) utricles. *Tmie* transcripts are absent in the entire sensory epithelium of *Gfi1*ᶜʳᵉ; *Tmie*ᶠ/⁻ utricle (**b'**) but are absent only in the Emx2-positive lateral region (**a–c**, black dotted region) of *Emx2*ᶜʳᵉ; *Tmie*ᶠ/⁻ utricles (**c'**). **d–f** FM1-43 staining in live utricles of *Tmie*⁺/⁻ (**d**, *n* = 5), *Gfi1*ᶜʳᵉ; *Tmie*ᶠ/⁻ (**e**, *n* = 3), and *Emx2*ᶜʳᵉ; *Tmie*ᶠ/⁻ (**f**, *n* = 4) inner ears at P0. FM1-43 is present in the entire sensory epithelium of *Tmie*⁺/⁻ (**d**), absent in *Gfi1*ᶜʳᵉ; *Tmie*ᶠ/⁻ (**e**) and absent only in the lateral region of *Emx2*ᶜʳᵉ; *Tmie*ᶠ/⁻ utricle (**f**). Outline of the utricular macula is demarcated with a white dotted line and the lateral region is marked with a yellow dotted line (**d–f**). **g–l** Utricles with dye tracing from the cerebellum (green) and brainstem (magenta) of *Tmie*ᶠ/⁻ (**g**, *n* = 2; **j**, *n* = 3), *Gfi1*ᶜʳᵉ; *Tmie*ᶠ/⁻ (**h**, *n* = 2; **k**, *n* = 3), and *Emx2*ᶜʳᵉ; *Tmie*ᶠ/⁻ (**i**, *n* = 3; **l**, *n* = 3) at E16.5 day (**g–i**) and P0 (**j–l**). Neuronal tracing from the cerebellum and the brainstem labeling the respective lateral and medial utricle are normal among all three genotypes at E16.5 and P0. Scale bar = 200 μm in **a**, applies to **b, c**; 100 μm in **a'**, applies to **b', c'**, and 100 μm in **d–l**.

of stimulation were correlated with a higher threshold change and no difference in thresholds in the negative direction of stimulation (Fig. 8e). Taken together, these VsEP results show that both *Emx2* cKO and *Tmie* cKO lose sensitivity from some directions of stimulation. Further, there is a gain in the remaining directions of stimulation in *Emx2* cKO that is not present in the *Tmie* cKO.

## Discussion

The results reported here indicate that Emx2 mediates bidirectional selectivity of the otolith organs. *Emx2* expression in nascent HCs is sufficient to mediate hair bundle reversal from the default pattern based on HC-specific KO of *Emx2* results (Fig. 3). In contrast, knocking out *Emx2* in HC alone only has a minor effect on neuronal innervation

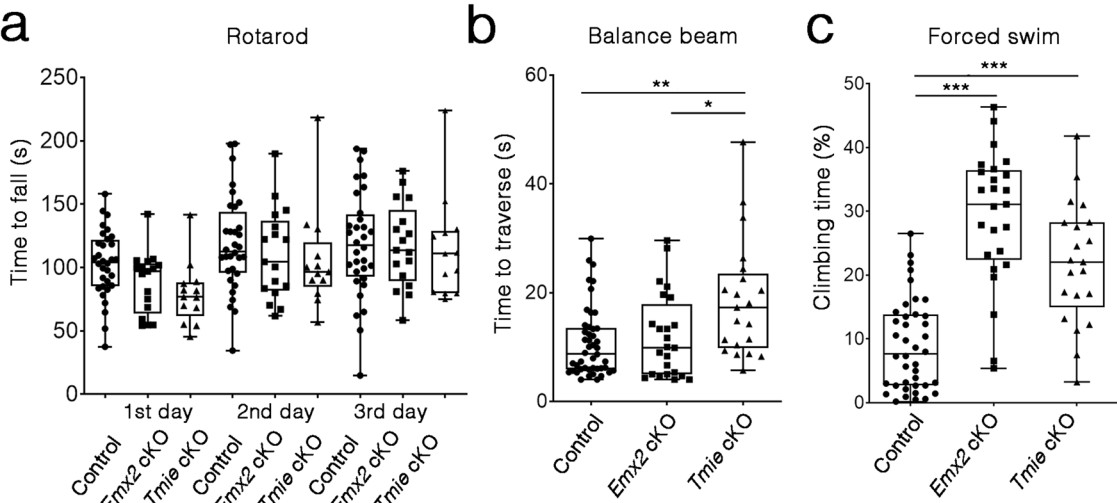

**Fig. 7 | Panic swimming for both *Emx2* cKO and *Tmie* cKO but *Tmie* cKO not *Emx2* cKO showed deficits on balance beam. a** Quantification of time for mice to fall off a rotating rod, which accelerated from 5 to 40 r.p.m. over a 5-min period. The performance of *Emx2* cKO and *Tmie* cKO are similar to controls on all three days of the trial (control, *n* = 32; *Emx2* cKO *n* = 17; *Tmie* cKO *n* = 13). **b** Quantification of time required for control, *Emx2* cKO, and *Tmie* cKO to traverse an 80-cm distance on a 6-mm-wide beam. *Tmie* cKO mutants took longer to cross the beam but *Emx2* cKO were normal, compared to controls (control *n* = 46; *Emx2* cKO *n* = 24; *Tmie* cKO

*n* = 21, *Emx2* cKO vs *Tmie* cKO *\*p* = 0.02, control vs *Tmie* cKO *\*\*p* = 0.0013). **c** Percentages of time mice spent climbing during the 1-min period in water. *Emx2* cKO and *Tmie* cKO spent more time climbing in water, when compared to controls (control *n* = 40; *Emx2* cKO *n* = 25; *Tmie* cKO *n* = 21; control vs *Emx2* cKO *\*\*\*p* = 2.64E−13 and *Tmie* cKO *\*\*\*p* = 1.72E−07). In boxplots, boxes represent the interquartile range (IQR), and the thick lines inside show the median. Whiskers denote the lowest and highest values within 1.5 times the IQR. The one-way ANOVA with multiple comparisons was applied. Source data are provided as a Source data file.

pattern obscuring the clear segregation border between the two types of vestibular neurons that normally innervate the maculae (Fig. 3). This minor effect suggests that Emx2 in HCs is not the sole source responsible for establishing neuronal selectivity, and its expression in supporting cells is most likely required as well. Notably, ectopic expression of *Emx2* in supporting cells, though not as potent as ectopic *Emx2* in HCs, was effective in disrupting the neuronal segregation (Fig. 5). Vestibular fibers entering the maculae begin to be segregated at around E14.5[16], at which time most HCs have not formed yet[30] but the regional expression of *Emx2* in the sensory epithelium is already well-established[21]. Together, these results support the hypothesis that *Emx2* within the sensory epithelium/supporting cells guides the initial spatial segregation of the two types of afferents, namely those that project to the cerebellum versus brainstem. This segregation is then refined by *Emx2* expression in HCs as they develop. Thus, unlike the requirement of Emx2 in mediating hair bundle orientation, source of *Emx2* in HCs and most likely supporting cells are required for establishing the proper neuronal innervation pattern in the maculae.

Our results indicate that Emx2 has a cell-autonomous effect on hair bundle orientation in HCs, whereas its role in neuronal selectivity is less clear. In *Emx2* KO, neurons that project to the cerebellum failed to reach the sensory epithelium that normally expresses Emx2, and this region became populated by nerve fibers that project to the brainstem (Fig. 2). Even though ectopic expression of *Emx2* in HCs or supporting cells alone was not sufficient to inhibit brainstem neurons, it is plausible that the endogenous Emx2-positive sensory epithelium, HCs and supporting cells collectively, has a direct or indirect effect in excluding the brainstem neurons.

While the downstream effector(s) of Emx2 for both of its functions in the otolith organs has not been identified, it is known that post-translational regulation of Gpr156, a G-protein coupled receptor, by Emx2 is required for hair bundle orientation establishment[31]. Thus far, there is no direct evidence that implicates Gpr156 in the neuronal-selectivity function of Emx2 and Gpr156 is only expressed in HCs and not supporting cells[31]. Nevertheless, our results indicate that the timing and qualitative requirement of Emx2 in bundle orientation and neuronal selectivity are different. Ectopic expression of *Emx2* in the

maculae at E15.5 has a robust effect on disrupting the neuronal innervation across the utricle but only a minor effect on reversing hair bundle orientation in the medial and no effect in the lateral utricle. Thus, even if Emx2 transcriptionally regulates hair bundle orientation and neuronal selection via a common mediator, the temporal requirement of this mediator is different.

Multiple lines of evidence suggest that VsEP is generated by neurons that innervate the striola. The phase-locking and transient firing properties of the neurons innervating the striola are postulated to be the primary candidate in generating a synchronized, reproducible action potential of the VsEP elicited by the jerk stimuli[32,33]. Furthermore, the *Cyp26b1* cKO lack the striola, and only remnant VsEP was detectable in these mutants[5]. Both *Emx2* cKO and *Tmie* cKO mutants showed a relatively incremental reduction in VsEP amplitudes compared to the *Cyp26b1* cKO. We attributed this small reduction in VsEP response of the two LPR mutants to a compromised striola function in the saccule and not the utricle because Emx2 is partially expressed in the striola of the saccule and not the utricle (Fig. 1b)[13]. Thus, the striolar function in the utricle should not be affected in either mutant. Despite the small changes in VsEP, the two LPR mutants showed an orientation difference in VsEP response validating the hair bundle arrangement differences between the two mutants (Fig. 8b).

In principle, a stimulus of a given direction will generate two concomitant signals—excitatory and inhibitory—at the level of the maculae, which are in turn transmitted to the vestibular nuclei in the brainstem as well as the uvula and nodulus of the cerebellum (Figs. 1b and 8b). While there are known reciprocal connections between the brainstem vestibular nuclei and vestibular-associated nuclei in the cerebellum[34], it is unknown whether and how signals from the Emx2-positive and Emx2-negative regions of the maculae are integrated in the brain. Recent evidence indicates that primary inputs from the vestibular organs (include afferents innervating the Emx2-positive region) and secondary inputs from the vestibular nuclei (include receiving inputs from afferents innervating the Emx2-negative region) both converge on different subtypes of unipolar brush cells in the cerebellum[35]. Since unipolar brush cells normally converge to the

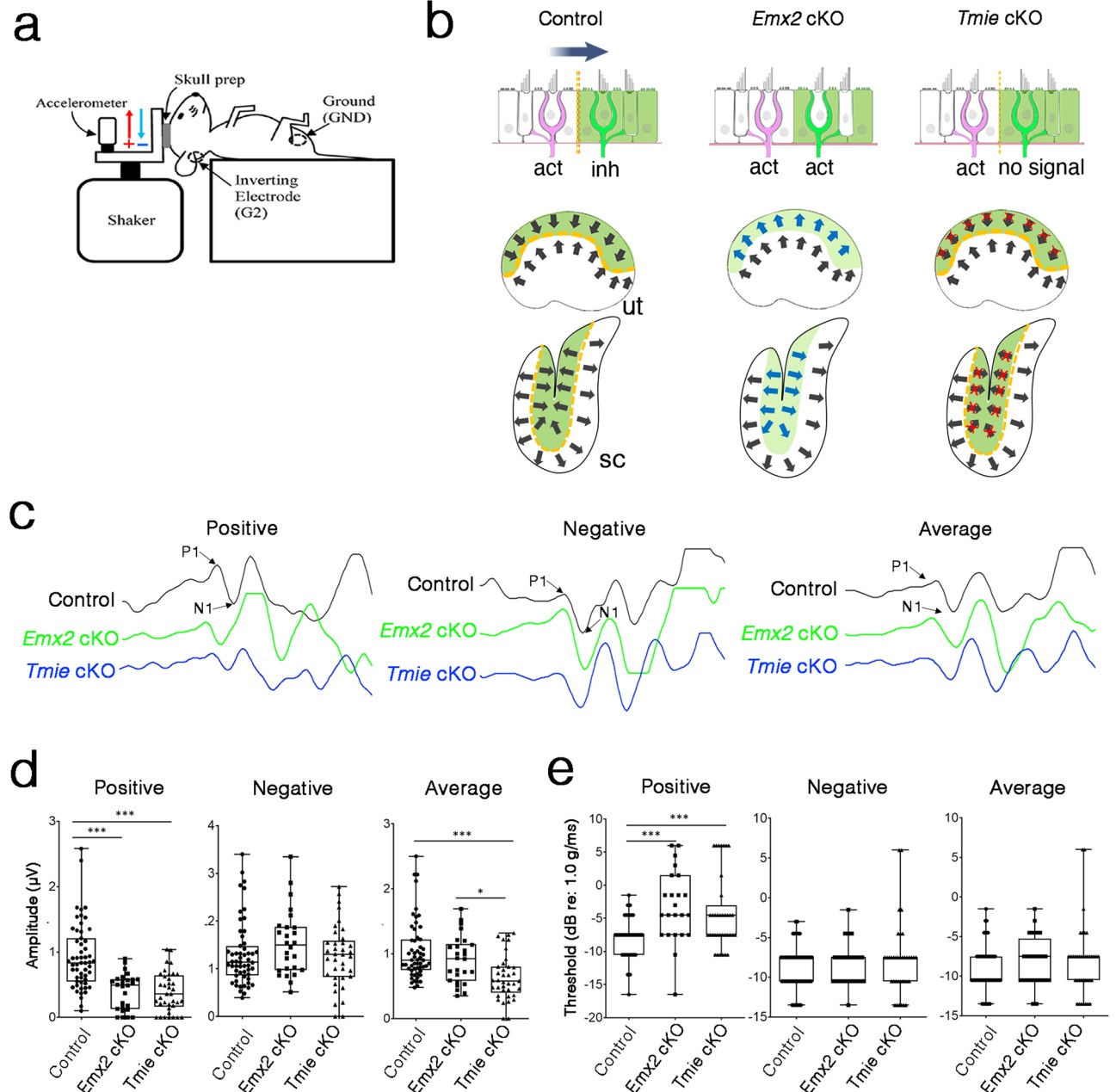

**Fig. 8 | Reduced amplitudes and increased thresholds of VsEP in *Emx2* cKO and *Tmie* cKO mice. a** Schematic diagram of the apparatus for VsEP measurement (Reprinted from Jones, T. A. & Jones, S. A. Short latency compound action potentials from mammalian gravity receptor organs. *Hear. Res.* **136**, 75–85 (1999). Copyright (1999), with permission from Elsevier). Red arrow indicates the direction of positive stimulation towards the nose, and the blue arrow indicates the direction of negative stimulation towards the occipit. **b** Hair bundle orientation of the maculae and neural responses to a directional stimulus (large gray arrow) in the utricle of controls, *Emx2* cKO and *Tmie* cKO. **c** VsEP waveforms at maximal jerk stimulus (+6 dB) from positive and negative directions of stimulation as well as the average of the two stimulus directions for controls (black), *Emx2* cKO (green) and *Tmie* cKO (blue). **d** Summary of VsEP amplitudes measured at maximal jerk stimulus (+6 dB) of positive and negative directions of stimulation, and average of the two stimulus directions. Both mutants show lower amplitudes in positive direction of stimulation

(Control vs *Emx2* cKO \*\*\**p* = 4.04E-06 and Tmie cKO \*\*\**p* = 8.15E-08), but no difference in negative direction of stimulation, compared to controls. Only *Tmie* cKO and not *Emx2* cKO show a lower average amplitude compared to controls (*Emx2* cKO vs *Tmie* cKO \**p* = 0.03, Control vs *Tmie* cKO \*\*\**p* = 1.36E-05). **e** Summary of thresholds for VsEP determined by various jerk magnitudes. Both *Emx2* cKO and *Tmie* cKO mice show higher thresholds in the positive direction of stimulus (Control vs *Emx2* cKO \*\*\**p* = 1.02E−05 and *Tmie* cKO \*\*\**p* = 1.64E−04) but no significant difference in negative direction of stimulus or average of the two directions stimuli, compared to controls. Control *n* = 55; *Emx2* cKO *n* = 26; *Tmie* cKO *n* = 35. In box-plots, boxes represent the interquartile range (IQR), and the thick lines inside show the median. Whiskers denote the lowest and highest values within 1.5 times the IQR. The one-way ANOVA with multiple comparisons was applied. Source data are provided as a Source data file.

granule cells[36], this could be one of the locations where inputs from Emx2-positive and negative regions are combined and integrated.

Without more concrete validation of these possibilities, it is difficult to postulate the underlying cause for the small differences in vestibular deficits observed between the two LPR mutants. While *Tmie*

cKO mutants show more difficulty in traversing the balance beam, *Emx2* cKO mutants spent more time trying to climb out of the water during the swimming task (Fig. 7). The bundle orientation and physiological differences of these mutants will provide a valuable tool for identifying the cellular location(s) where the normal integration of

these macular inputs to the cerebellum and brainstem takes place. Take together these corroborative behavioral phenotypes and the lack of obvious *Emx2* expression along the primary neuronal circuitry responsible for detecting directional sensitivity such as the vestibular ganglion, the cerebellum and vestibular nuclei in the brainstem[37–39] (Allenbrain atlas, https://portal.brain-map.org/explore/overview?gclid=Cj0KCQjw54iXBhCXARIsADWpsG9QQ6yh36gCEmLK1OG-onQSIjfKUaP8xxaQv5FYS8DkPDqALThqOUwaAj-_EALw_wcB), the most plausible explanation of our results is that the main source of Emx2 that mediates this bidirectional sensitivity lies in the restricted expression of Emx2 in the maculae.

In addition to its macula input, the cerebellar uvula and nodulus receive additional sensory inputs[34]. The integration of maculae, cristae, visual and other sensory information is thought to underlie the brain's representation of spatial orientation[40,41]. We speculate that the observed panic behavior in water of *Emx2* cKO and *Tmie* cKO are due to altered vestibular–extravestibular integration at the level of the vestibular cerebellum resulting from the lack of a functional LPR. Interestingly, mutants without a striola, for example *Cyp26b1* cKO, have no measurable VsEP yet do not demonstrate climbing behavior observed here or other detectable impairments during swimming[5]. Thus, these differential requirements of specific vestibular components in mediating certain behavior point to a regional division of labor mechanism in sensory processing at the periphery organs.

## Methods

### Mice

All animal experiments were conducted under the approved NIH animal protocols at the NIH (#1212-17), University of Nebraska−Lincoln, Johns Hopkins University and according to NIH animal user guidelines. Mice were housed on individually ventilated cage (IVC) racks in a conventional vivarium on a 12 h/12 h light/dark cycle with temperature and humidity maintained within the following ranges, 21–24 °C and 30–70%, respectively. Mice of both sexes were used in all the experiments conducted. The *Rosa26^Emx2* mouse was generated as described previously[13]. Other published mouse strains that were used in the study and their source are as follows: *Emx2^Cre*, maintained in a CD-1 background, was from Shinichi Aizawa at RIKEN Center for Developmental Biology (RRID:IMSR_RBRC02272)[42]; *Emx2^flox* (*Emx2^F*)[38], maintained in a mixed C57BL/6J/129 background was from Andreas Zembrzycki at the Salk Institute; *Emx2^+/−*, maintained in a mixed C57BL/6J and CD1 background, was from Peter Gruss at the Max-Planck Institute (RRID:IMSR_EM:00065)[39]; *Gfi1^Cre*, maintained in a CD-1 background, was from Lin Gan at Augusta University, Georgia (PRID:MGI:4430258)[22]; *Gfap^cre* (RRID: IRSR_JAX:004600)[23], *Plp^CreER* (RRID: IMSR_JAX:005975)[25] and *Rosa26^tdTomato* (RRID:IMSR_JAX:007914)[43] were from Jackson Laboratory; *Sox2^CreER* was from Konrad Hochedlinger at Harvard University (RRID:IMSR_JAX:017593)[44]; and *Tmie^+/−* and *Tmie^flox* (*Tmie^f*), maintained in a C57BL6J/129 background, were from Ulrich Müller at Johns Hopkins University[26]. The littermates of *Emx2* cKO and *Tmie* cKO are both in a mixed CD1/C57BL/6J/129 genetic background.

### Tamoxifen treatment

A stock solution of 1 ml of corn oil from grocery store containing 30 mg tamoxifen (T5648, Sigma Aldrich, St. Louis, MO) and 0.2 mg β-estradiol (20 mg/ml of ethanol; E8875, Sigma Aldrich, St. Louis, MO) to avoid premature abortion of fetuses due to tamoxifen was prepared. On designated gestation days at noon, pregnant females were gavaged with the stock solution of tamoxifen at 1 mg/10 g body weight. The morning of a vaginal plug was considered as embryonic day 0.5.

### Lipophilic dye tracing

Neuronal tracing of vestibular neurons was conducted as described[45]. Briefly, the head of perfused mouse embryos was hemi-sectioned, and a piece of filter paper soaked with lipophilic dye, NeuroVue® Maroon,

was inserted into the cerebellum at the presumptive uvula and nodulus position. Then, a filter paper soaked with NeuroVue® Red was inserted into the brainstem for retrograde tracing of fibers into the vestibular sensory organs.

To quantify the extent of neuronal segregation in utricles, we analyzed the sensory epithelium of the utricle using an ImageJ macro. Briefly, an 8-bit rectangle was drawn on the sensory epithelium of a z-stack image taken at ×40 magnification, in which the relative intensities of red and green signals (y axis) were scored along the x axis from the lateral to the medial edge of the sensory epithelium. The length of the sensory epithelium selected (lateral to medial) ranges between 200 and 290 μm in distance and the lateral one-third was considered as the lateral extrastriolar region (area #1, 4, or 7), whereas the remaining two-thirds were considered as the medial regions (area #2, 3, 5, 6, 8 or 9). Within each region, relative dye intensity values for each color (obtained by the (intensity value−background value)/maximum intensity value) from approximately 583- to 841-pixel lines (each pixel line = 0.346 μm) were converted into binary scores of either 1 (green) or 0 (red). Then, the relative cb signal obtained by cb signals divided by total signals from cerebellum and brainstem (cb + bs), were calculated within each region. To be able to compare the extent of neuronal disruption among different mutants, we selected four representative control samples for intensity quantification, one *Emx2^F/+*, one *Emx2^+/−*, and two *Rosa^Emx2/+* utricles, and all mutant analyses were compared against these four controls.

### Wholemount immunostaining

Hemi-sectioned embryonic heads at E16.5 were fixed with 4% paraformaldehyde in PBS at 4 °C overnight. The utricle attached to the anterior crista and lateral crista were dissected as whole, washed three times with PBS and then put into blocking solution with PBS containing 4 % donkey serum and 0.2 % Triton-X for 2 h. Then, tissues were incubated with primary antibodies overnight at 4 °C, followed by incubation with secondary antibodies (Goat anti-mouse IgG Alexa Fluor 488, Cat #: A11029 and Goat anti-rat IgG Alexa Flour 568, Cat #: A11077, Thermo Fisher Scientific, Waltham, MA) at 1:1000 dilution for 1 h at room temperature after repeated washing with PBS. Samples were washed extensively again before mounting in ProLong Gold Antifade (Thermo Fisher Scientific, Waltham, MA) and imaged with a Zeiss LSM780 confocal microscope. Primary antibodies of anti-βII spectrin (1:500; Cat #: 612562, BD Biosciences, San Jose, CA) and rat anti-tdTomato 16D7 (1:200; Cat #: EST203, Kerafast, Boston, MA) were used.

### Immunohistochemical staining of cryosections

Cryosections of mouse utricles at 16 μm thickness was processed for anti-immunostaining as described for wholemount immunostaining. Primary antibodies used were as follows: mouse anti-beta-III tubulin (Tuj1) at 1:500 dilution (Cat #: MAB1195, R&D systems, Minneapolis, MN), rabbit anti-cleaved caspase-3 D175 (activated form of caspase-3) at 1:300 dilution (Cat #: 9661, Cell Signaling, Danvers, MA) and rabbit polyclonal anti-Myosin7a at 1:1000 dilution (Cat #: 25-6790, Proteus Bioscience, Ramona, CA), with secondary antibodies (Goat anti-mouse IgG Alexa Fluor 488, Cat #: A11029 and Goat anti-rabbit IgG Alexa Flour 568, Cat #: A21069, Thermo Fisher Scientific, Waltham, MA) at 1:1000 dilution.

### BaseScope

To determine the efficiency of recombination across the *loxP* sites in exon 5 (150 bp) of the *Tmie floxed* allele, we conducted RNA in situ hybridization using BaseScope™ detection reagents v2 (323910, Advanced Cell Diagnostics), designed to detect a small sequence of RNA. BaseScope probes used in this study were a "ZZ" antisense probe targeted against the *Tmie* exon 5 sequences, which should be absent in both the null and *loxP* allele (842341, Advanced Cell Diagnostics), and a

"ZZ" negative control probe (701021, Advanced Cell Diagnostics). In situ hybridization experiment was repeated twice.

## FM1-43 staining

Fresh utricles were dissected in HBSS (Cat# 14025, Gibco) and mounted on 35 mm glass bottom dishes (P35-0-10-C, MatTek), which were pre-coated with Cell-tak (354240, Corning) for 3 h or overnight and dried at room temperature before use. A 5-µM FM1-43 (T35356, Invitrogen) solution, diluted from a stock solution of 2 mM in methanol, was applied quickly to the mounted tissues for 10 s, followed by washing three times with HBSS and imaged at the confocal microscope after 5 min at room temperature.

## Vestibular testing

All vestibular testing were conducted on mice between 4 and 10 months of age. Measurements of *Emx2* cKO and *Tmie* cKO were compared to littermate controls measured on the same day with various available genotypes such as cre$^{+/-}$; *Flox*$^{+/-}$, null$^{+/-}$; *Flox*$^{+/-}$, and *Flox*$^{+/-}$. Controls for vestibular testing analyses were combined controls from both *Emx2* cKO and *Tmie* cKO, unless indicated otherwise.

**Rotarod test.** Each mouse was placed on a motorized rotating rod (ROTA ROD, Panlab, Harvard Apparatus) that gradually accelerated from 5 to 40 rpm in 5 min, and the time required for the mouse to fall off the rotarod was scored. Each mouse underwent tests for three consecutive days with 5 trials per day. First day was considered as the training day. Averaged score for each day was processed for statistical analysis.

**Balance beam test.** Mice were subjected to traversing on two different balance beams: a 70-cm long beam that was 12-mm wide and an 80-cm long beam that was 6-mm wide. Beams were positioned at 80 cm above ground with a dark box on one end and a harness underneath to catch any mouse that might fall. The time it took a mouse to reach the dark box from the opposite end of the beam was recorded. Mice were scored as failed when they did not reach the endpoint within 2 min. On the first day of the trial, some mice including control mice failed. By the second day of the trial, only one *Emx2* cKO and one *Tmie* cKO failed to reach the end point. Only times recorded on the second day were quantified, not including the two mice that failed to reach the end point within 2 min.

**Forced swim test.** Mice with an intact vestibular system swim at the water surface whereas those with abnormal vestibular function either swim in circles or rolling to one side or are unable to swim at the water surface[46,47]. Swimming ability was tested in a round plastic tub of 20 cm diameter filled with water up to 15 cm of height at a temperature of 24–26 °C. Each mouse was held by its tail and released into the tub from a height of 5 cm, and its swimming ability was recorded for 1 min. The duration a mouse spent trying to climb out of the water (vertical position) was measured by analyzing recorded videos using the ForcedSwimScan™ 2.0 software (Clever Sys Inc). Schematic of the forced swim set up and the climbing behavior of mice in water can be found in https://www.semanticscholar.org/paper/High-speed-video-analysis-of-laboratory-rats-in-Nie-Ishii/7070f70210f31b5e43832a70e0e0c245a43f735c.

**Open-field test.** The open field test under red light was performed to measure hyperactivity in mice. After adapting in the testing room for 10 min, individual mice were placed in an open arena of a 30 cm by 30 cm plexiglass chamber and behavior of each mouse was recorded using an overhead camera for 30 min at 30 frames/s (EverFocus EQ700, Taipei, Taiwan). Recordings of mouse behavior were analyzed using Topscan software (3.0, Clever Sys Inc.), and distance traveled for each mouse within a 5 min time period was used to determine hyperactivity.

**Angular vestibular ocular reflex and optokinetic reflex.** To quantify the eye movements evoked by the aVOR, the turntable was sinusoidally rotated at frequencies of 0.2, 0.4, 0.8, 1, and 2 Hz with peak velocities of ±16°/s in both light and dark. To quantify the eye movements evoked by the OKR, the turntable remained stationary while the visual surround (vertical black and white stripes, visual angle width of 5°) was sinusoidally rotated at frequencies of 0.2, 0.4, 0.8, 1, and 2 Hz with peak velocities of ±16°/s. Recorded eye, head, and visual surround movement signals were low-passed filtered at 125 Hz and sampled at 1 kHz. Eye position data were differentiated to obtain velocity traces. Cycles of data with quick phases were excluded from the analysis. Least-square optimization determined the aVOR and OKR gains, and phases plotted as mean ± standard error of the mean (SEM) against all frequencies for all mice.

**VsEP measurements.** VsEP recordings were conducted according to methods previously described[48,49]. Briefly, an anesthetized mouse was placed supine on a temperature-controlled heating pad to maintain core body temperature at 37 °C, and its head was attached to a mechanical shaker (Labworks Inc. ET 132-2, Costa Mesa, CA, USA) using a custom noninvasive head clip. The mechanical shaker was used to deliver linear jerk pulses to the head along the naso-occipital axis. Subcutaneous recording electrodes were placed at the nuchal crest (noninverting) and behind the left pinna (inverting). A ground electrode was placed at the right hip. Two hundred and fifty-six linear acceleration ramps with a 2-ms duration were delivered at a rate of 17 pulses per sec in either the positive (occipital towards nasal) or negative (nasal towards occipital) direction to stimulate the gravity receptor organs in the inner ear (saccule and utricle). Stimulus amplitude ranged from +6 dB to −18 dB in 3 dB steps, re: 1.0 g/ms. Signal averaging was used to produce response waveforms at each stimulus level for positive and negative directions as well as an average of both stimulus directions. The resulting waveform consisted of three positive and negative peaks where the earliest response peak (P1−N1) reflects the vestibular portion of the eighth nerve innervating the gravity receptor organs[50]. Amplitudes representing the peak-to-peak magnitudes between the first positive (P1) and negative (N1) response peaks were measured in micro-volts (µV). The averaged VsEP response represents averaging a total of 512 stimuli from both directions. Threshold was defined as the stimulus intensity (measured in dB, re: 1.0 g/ms) half-way between the minimum intensity producing a response and the maximum intensity failing to produce a response[51].

## Statistical analysis

All statistics were conducted using Prism 7 (GraphPad Inc.) or R version 4.1.3. One-way analysis of variance (ANOVA), followed by Tukey's multiple comparison test or two-way ANOVA with post hoc Bonferroni's test were conducted as indicated. Data are shown as average ± SEM unless indicated otherwise. Data distribution was assumed to be normal, but the distribution was not tested formally.

Multiple linear regression was used to analyze the dye labeling results. To increase statistical power, we generated a new group variable "Region" by coding areas # 1, 4, and 7 as lateral regions, areas # 2, 5, and 8 as intermed_medial regions, and areas # 3, 6 and 9 as medial regions (refer to schematic in Fig. 3e). The new group variable was used in the analysis. A regression model was developed to examine the effect of region and genotype on the relative cb signals. To justify the use of linear regression model, we tested the assumptions with regression diagnostics plots and gvlma (Global Validation of Linear Models Assumptions)[52] package in R. Sample size estimation was conducted with a priori power analysis tool, G*Power version 3.1.9.6[53]. Power analysis showed that we had sufficient sample size for multiple linear regression. Outliers were identified with the diagnostics plots. The outlier exclusion was on two-fold basis: First, the outliers need to lie outside 1.5*IQR (Interquartile Range), as visualized in the box-

whisker plot. Second, the outliers were influential datapoints identified by Cook's distance, which were at least greater than four times the mean. After the removal of five outliers in $Sox2^{CreER}$; $Rosa^{Emx2}$ data and three outliers in $Gfi1^{Cre}$; $Rosa^{Emx2}$ and $Plp^{CreER}$; $Rosa^{Emx2}$ data, as well as the addition of an interaction term Region*Genotype, all the assumptions were satisfied, which include normal distribution and independence of residuals, homoscedasticity, and linear relationship between dependent and independent variables. The model fit has been improved substantially. We have improved the adjusted $R^2$ from 0.45 to 0.59 and from 0.47 to 0.53 for the above-mentioned data.

Further comparisons between genotypes or regions were made using the Kruskal–Wallis rank-sum test. Dunn's pairwise comparisons tests were used for post hoc procedure on each pair of groups. Benjamini–Hochberg multiple testing correction was used to control the false discovery rate. The coefficient estimates are given with 95% confidence intervals (CI). $p < 0.05$ (two-sided) is considered statistically significant. For the $Gfi1^{Cre}$; $Tmie^{F/-}$ and $Emx2^{Cre}$; $Tmie^{F/-}$ data, only Kruskal–Wallis rank-sum test and Dunn's pairwise comparisons were conducted because the assumptions for linear regression model were not met even after removal of outliers and the addition of an interaction term.

## Reporting summary

Further information on research design is available in the Nature Research Reporting Summary linked to this article.

## Data availability

All relevant data and source data are included in this article and Supplementary Information. Source data are provided with this paper.

## Code availability

The ImageJ macro used to quantify the intensity of lipophilic dye in utricles can be found in https://github.com/Wulab2022.

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

## Acknowledgements

We would like to thank Ulrich Müller at Johns Hopkins University for the $Tmie^{+/-}$ and $Tmie^{flox}$ ($Tmie^{f}$) mice, Andreas Zembrzycki for the $Emx2^{F}$ mice, and Amelia Baumgardner for assistance with VsEP data analysis. We also thank Dr. Yogita Chudasama, Kevin Cravedi, and Alice Graham in the Mouse Behavioral Core at NIMH for the use of the facility and assistance, and NIH Medical Arts for drawing and permission to use the schematic diagrams. We are also grateful to Dr. Thomas Friedman and Dr. Katie Kindt and members of our laboratory for critical reading of the manuscript. This study is supported by NIDCD Intramural program (1ZIADC000021) awarded to D.K.W. and NIDCD Mouse auditory testing core facility (ZIC DC000080), National Institutes of Health funding (UF1NS111695, R01DC018304, RDC002390, and R01DC018061) awarded to K.E.C, and National Institute of Aging funding (R01 AG060504) awarded to B.F.

## Author contributions

Y.J. and D.K.W. conceived, designed, and wrote the manuscript. Y.J., Y.T., and T.W., performed the experiments, analyzed the data, and contributed to writing. T.J., M.E.C., and E.D.T. performed the experiments, analyzed the data. S.O. wrote the ImageJ macro, T.F. analyzed the data. B.F., S.M.J., and K.E.C. designed, analyzed the data, and contributed to writing the manuscript. H.C. conducted statistical analysis and contributed to writing the manuscript.

## Funding

## Competing interests

The authors declare no competing interests.
