## [Peer Review File · Nature Communications]

Function of bidirectional sensitivity in the otolith organs established by transcription factor Emx2REVIEWER COMMENTS

Reviewer #1 (Remarks to the Author):

This paper reports interesting and compelling data that enhance our understanding of how information is encoded by the vestibular system. We use our vestibular system to sense head position as we move through the world. Despite its critical importance, we still lack basic information about the logic of circuit organization and function as information flows from the peripheral sensory organs (the maculae and cristae) to the central nervous system (i.e. the vestibular nuclei and the cerebellum). For example, in the utricular and saccular maculae, hair cells flip their orientation along a well-defined line of polarity reversal (LPR), but the functional significance of this striking organization is not known. This group previously showed that the transcription factor *Emx2* is expressed on one side of the LPR and required for flipping the orientation of hair cells in that region. Here, they use mouse genetics and dye labeling to show how hair cells on either side of the LPR are innervated and how this organization affects vestibular responses and behavior. By altering where they implant dye, they show that vestibular ganglion neurons (VGNs) that innervate the lateral utricle and inner saccule send their central axons to the cerebellum, whereas those that innervate the other side of the LPR project instead to the brainstem, with little overlap (Fig. 1, 3). In *Emx2* knock-out mice, there is reduced innervation by the cerebellum-projecting VGNs and expanded innervation by brainstem-projecting VGNs, a phenotype that is accompanied by increased cell death in the ganglion (Fig. 2, Supp Fig 1-3). Then, they show that this segregation is less precise when *Emx2* is removed from hair cells (Fig. 3, Supp Fig 4) but that supporting cells likely contribute, since the neuronal phenotype is milder than the full knockout and can be separated from effects on hair cell bundle orientation (Fig. 4-5, Supp Fig 5). For instance, ectopic expression of *Emx2* using PLP-CreER is sufficient to disrupt the innervation pattern even in the absence of changes to hair cell bundle orientation (Fig. 5, Supp Fig 6). Finally, the authors explore the functional implications of this organization both by preventing mechanotransduction in hair cells that synapse with cerebellar-projecting VGNs (Fig. 6) and by flipping hair cell orientation on that same side. This is a valuable comparison as in one case, one line of hair cell communication is silenced (*Tmie* CKO) and in the other case, hair cells that would normally be inhibited by a stimulus are now activated (*Emx2* CKO). Neither mutant shows obvious vestibular deficits and they can even perform the classic rotorod test quite well. However, the *Tmie* CKOs are slower to make their way across a narrow balance beam, and both strains show changes in swimming behavior (Fig. 7, Supp Fig 7, 8). Finally, vestibular evoked potentials (VsEPs) show predicted changes, consistent with the idea that this physiological measure reflects the activity of neurons projecting to the striola, which straddles the LPR (Fig. 8).

Overall, this is a thorough, high quality study. The findings are robustly quantified and supported with multiple lines of evidence. I have only a few minor suggestions for improvement.

1. While the CKO mice provide strong evidence that *Emx2* effects on neuronal innervation arise from the sensory epithelium, I did wonder whether *Emx2* is expressed elsewhere, such as the brainstem or cerebellum. I did a few literature searches and couldn't quickly come up with the information. It would be helpful to add a bit more to help others to interpret the findings.
2. The Basescope in situ hybridization results are convincing, but it is a bit hard to see the signal. It would be good to explain why this method was used (presumably because of the small size of the deleted exon) for readers who might wonder.
3. The open field test results are mentioned but the data are not shown. It would be good to add as a supplementary figure.
4. Please note whether the *Sox2-CreER;Rosa-Emx2* mice show any behavioral deficits.
5. I appreciate how challenging it is to quantify dye labeling results, since one cannot control for the degree of dye uptake. The method used makes sense overall and matches what is apparent qualitatively. However, I was a bit confused by some of the results, such as the lack of significant difference between control and *Sox2CreER;Rosa-Emx2* in area #2 (Supp Fig. 5). I believe the authors but would just like some explanation to help me understand the possible sources of variation. The

data would be better presented as raw data points in box and whisker plots rather than bar graphs with error bars. Also, I would be interested to see the single channel images for at least one of the dyes. It would also be good to get a sense of any variation in the degree of dye labeling. For instance, if possible, single channel and merged images for all four controls could be shown, especially since it seems that the same controls are used for all of the comparisons (this does seem less than ideal). Finally, related to the dye labeling, I wondered whether the region of overlap indicated by asterisks in Fig 1 would show up as quantitatively similar to the degree of overlap that occurs in the mutants. I believe that that region is not included in any of the sampled areas, but if it is, please note. Additionally, it would be good to comment briefly on how the lack of segregation in this region in control animals might affect the overall model for how Emx2 acts in developing sensory cells to influence the final connectivity of the VGNs.

6. Most of the dye labeling is done at E16.5, but it isn't clear why this embryonic timepoint was selected. In general, a bit more information about what is known about VGN innervation patterns and their development could be added to the introduction. On a related note, a lot happens between E16.5 and four months of age, the earliest timepoint for the functional assays. It would be good to examine at least one mature timepoint to see if the pattern seen at E16.5 is stable. In particular, I wondered whether some of the observed lack of segregation could be caused by retention of extra branches that might subsequently be pruned during subsequent synaptic refinement events.

Reviewer #2 (Remarks to the Author):

The manuscript by Ji et al. investigates the role of the transcription factor Emx2 in innervation patterns and bidirectional sensitivity in the mouse utricular and saccular maculae. Previous studies have shown that Emx2 expression demarcates the LPR and generates mirror-image hair cell orientation across the LPR. Combining conditional Emx2 loss- and gain-of-function mutants, retrograde tracing and behavioral analysis, the current study showed that Emx2 acts in both hair cells (HC) and supporting cells (SC) in the sensory maculae to guide vestibular afferent projection to the cerebellum, while the Emx2-negative sensory domain directs afferent projection to the brainstem. Emx2-directed afferent projection is likely distinct from its cell-autonomous role in reversing hair cell orientation. Moreover, by genetic ablation of hair cell mechanotransduction, it was shown that the afferent projection patterns form independently of HC activity. Finally, HC and neuronal defects in bidirectional sensitivity lead to subtle but unique behavioral defects.

Although the downstream effectors of Emx2 remain to be identified, this is a comprehensive study that provides significant new insights into peripheral innervation of balance organs, differentiating the contributions of Emx2, HC bi-orientation and HC activity. The data presented are largely convincing; however, I have the following suggestions to further improve rigor:

1. Quantitative analysis of dye tracing data is key in establishing the role of Emx2 in vestibular innervation. However, the statistical analysis used may not be the most appropriate. Specifically, without normality tests, one-way ANOVA is not valid for the small sample sizes analyzed in all the tracing experiments (n=2-6). A non-parametric test that makes no assumption about the distribution of data, such as the Kolmogorov-Smirnov test, may be better at revealing regional innervation differences between different genotypes. This would apply to data presented in Figures 3-6.

2. Several of the Cre lines used are also expressed in brain regions outside of inner ear, which could confound phenotype interpretation, in terms of afferent projection and behavioral defects. For example, Gfi1-Cre is expressed in the cerebellum, and the Cre allele results in Gfi haploinsufficiency. It would be important to use Gfi-Cre/+ samples as "control" groups in experiments involving this Cre driver.

3. Related to above, the Sox2creER and PIPcreER drivers are expressed in many types of glial cells

beside SCs in the sensory epithelia. For Emx2 overexpression experiments, a potential caveat is that Emx2 overexpression in various brain regions affected central projection patterns of vestibular afferents, contributing to the observed dye tracing defects. This is plausible because vestibular afferents are normally sensitive to Emx2-mediated cues from the sensory maculae, and should be at least discussed.

4. Dye tracing results in Figure 6 should be quantified.

Reviewer #3 (Remarks to the Author):

In this manuscript by Rae Ji et al the authors report the role that the transcription factor Emx2 plays in guiding the innervation of the mouse utricle and saccule and the functional consequences of disrupting afferent signaling to the cerebellum on vestibular function. Manuscript is centered upon an anatomical organization of the vestibular maculae that is a line of polarity reversal separating hair cells of opposite stereociliary bundle orientation. This line of reversal is correlated with the two major streams of afferent innervation projecting to either the cerebellum or the hindbrain. The authors are able to manipulate these parallel streams using two different mutants that selectively impact or silence cerebellar projections. This is an interesting addition to growing body of work on Emx2 that validates work in zebra fish showing that Emx2 coordinates innervation and bundle orientation in the neuromast (see ref#19 but also Lozano-Ortega et al 2018), and reinforces the contemporary view that information regarding linear acceleration is relayed from the maculae to the brain through parallel afferent paths.

Comments (in no significant order):

This is a solid technical effort with fantastic Neurovue tracing images that illustrate the utility of these tools far better than previous publications in which the dye labeling is heavily saturated

The combination of Emx2 CKO, Tmie CKO and vestibular behaviors demonstrate an overall focus on vestibular circuitry yet the abstract and introduction largely emphasize bundle polarity and bidirectional responses at the level of the hair cells. The authors should consider whether the unique aspects of their research are overshadowed by this structural organization. One could envision an abstract that began ... "The vestibular maculae are innervated by two parallel afferent projections to the hindbrain and cerebellum which are coordinated with the line of polarity reversal. This innervation is coordinated with bundle polarity but the significance is not known"

Writing styles differ between authors but one might argue that there is an unusually long summary of results in the last paragraph of the introduction

Line 96: The statement that "vestibular afferents that project to the cerebellum is replaced by neurons that projected to the brainstem in Emx2 knockouts" seems like an overstatement since these fibers still appear present (albeit reduced) in Fig2D.

Background strains for Tmie CKO and Emx2 CKO mice are not stated. As a result it is unclear whether it is appropriate for vestibular testing to combine control mice from these two different experimental crosses

Line 168: The following interpretation of Emx2 KO data "Together, these results suggest that normal Emx2 expression in the lateral utricle and inner saccule selects for neurons that project to the cerebellum and inhibits neurons that project to the brainstem." Is inconsistent with Emx2

overexpression data because neurons projecting to the brain stem are still present and therefore are not subject to 'inhibition'

Similarly in Discussion line 407 "it is equally likely that the exclusion of brainstem neurons is indirectly mediated by neurons that normally project to the cerebellum." It is hard to rationalize an exclusion model when the manuscript contains experimental conditions in which neurons overlap.

The data reported as a CB:BS ratio does not appear to be an actual ratio (in which case one would expect values greater than 1 in lateral regions). Instead these appears to report of percentage of green pixels (which would be fine). Minor related point, the methods describes the ratio of green:red whereas the figures are green:magenta

Lines 259-263 how is it that Plp1Cre is not appropriate to generate SC-specific Emx2 CKOs but is the optimal choice for inducing SC-specific Emx2 overexpression?

August 8,2022

Reviewer #1

1. While the CKO mice provide strong evidence that Emx2 effects on neuronal innervation arise from the sensory epithelium, I did wonder whether Emx2 is expressed elsewhere, such as the brainstem or cerebellum. I did a few literature searches and couldn't quickly come up with the information. It would be helpful to add a bit more to help others to interpret the findings.

Based on the in situ hybridization and *Emx2* cre reporter activity, we were not able to detect *Emx2* expression in the vestibular ganglion between E11.5 to E16.5. Additionally, *Emx2* was not reported to be expressed in the brainstem and cerebellum (Zembrzycki et al, eLife 2015; Pellegrini et al, Development, 1996;Lowenstein ED et al, FEBS J, 2022, Allen Brain atlas) in both embryonic and adult ages. Even though these are not exhaustive expression analyses of *Emx2*, we feel the most likely explanation for our results as they currently stand is that *Emx2* expression in the sensory epithelium is the primary mediator for establishing the cellular basis for the bidirectional sensitivity function described. This working hypothesis has been added in the Discussion (P.22, line 458).

2. The Basescope in situ hybridization results are convincing, but it is a bit hard to see the signal. It would be good to explain why this method was used (presumably because of the small size of the deleted exon) for readers who might wonder.

This information has been added. See P.26, line 550 in the revised manuscript.

3. The open field test results are mentioned but the data are not shown. It would be good to add as a supplementary figure.

This information has been added. See P.15, line 326 (Results) and P.28, line 607 (Materials and Methods) in the revised manuscript.

4. Please note whether the Sox2-CreER;Rosa-Emx2 mice show any behavioral deficits.

The *Sox2-creER; Rosa-Emx2* mutants showed severe phenotype in the brain and because of the in utero tamoxifen we have never tried to investigate their viability postnatally. However, the mildest gain-of-function mouse model with no apparent gross defects, *Gfi1-CreER;Rosa-Emx2*, shows severe behavioral deficits, and they die around P20. In this mutant, the hair bundles are reversed in all three cristae in addition to the loss of LPR in maculae (Jiang et al, 2017). This information has now been added to P13, line 281.

5. I appreciate how challenging it is to quantify dye labeling results, since one cannot control for the degree of dye uptake. The method used makes sense overall and matches what is apparent qualitatively. However, I was a bit confused by some of the results, such as the lack of significant difference between control and Sox2CreER;Rosa-Emx2 in area #2 (Supp Fig. 5). I believe the authors but would just like some explanation

to help me understand the possible sources of variation. The data would be better presented as raw data points in box and whisker plots rather than bar graphs with error bars.

As mentioned above, we have brought in the expertise of a statistician to help us reanalyze our dye-tracing results. We first used multiple linear regression to analyze the data presented in Figures 3-5 because it is a powerful tool to estimate the relationship between response and multiple explanatory variables. To increase statistical power, we generated a new group variable "Region" by coding area # 1, 4 and 7 as Lateral region, area # 2, 5 and 8 as Intermed_Medial region, and area # 3, 6 and 9 as Medial region. The new group variable was used in the analysis and a regression model was developed to examine the effect of region and genotype on the relative cb signal.

To calculate sample size for accurate estimation, we conducted a priori power analysis using G*Power version 3.1.9.6. From our analyses, the estimated R^2 would be about 0.6 for Figure 3, and 0.5 for Figures 4 and 5. The residual variance (defined as $1-R^2$) would be 0.4 for Figure 3, and 0.5 for Figures 4 and 5. The power analysis based on F test indicated the required sample size to achieve 70% power at a significance criterion of $\alpha = 0.05$ for linear regression analysis of Figure 3 data was $N = 80$, for Figures 4 and 5 data was $N = 105$. We concluded from this analysis that we have sufficient sample size for multiple linear regression analyses.

These regression diagnostics plots and validation testing showed that the assumptions of linear model were satisfied, which included normal distribution and independence of residuals, homoscedasticity, and linear relationship between dependent and independent variables. Then, we applied the nonparametric method Kruskal-Wallis rank sum test to compare the relative cb signal between different genotypes at different regions. Dunn's multiple comparisons tests were used for post hoc procedure on each pair of groups. These results are presented in box and whisker plots in Figure 3 to 5 and the statistical analyses were summarized in legend of Figures 3 to 5 and their respective Supplementary Tables 1 to 3.

Previously, using one-way ANOVA analyses, we concluded that: 1) *Emx2* cKO only has minor neuronal segregation defects at the border where the two types of neurons normally segregate, 2) gain of function of *Emx2* in the sensory epithelium at E13.5 or E15.5 showed similar neuronal perturbation phenotypes, and 3) gain of function of *Emx2* in sensory hair cells was more effective than gain of function of *Emx2* in supporting cells. These conclusions are still valid after applying the more appropriate statistical tools.

Also, I would be interested to see the single channel images for at least one of the dyes. It would also be good to get a sense of any variation in the degree of dye labeling. For instance, if possible, single channel and merged images for all four controls could be shown, especially since it seems that the same controls are used for all of the comparisons (this does seem less than ideal).

The impetus of comparing each mutant to the four combined controls allows us to compare effects of different mutant groups. The single channel images of the four controls are now illustrated in Supplementary Figure 4 of the revised manuscript. See P.10, line 196.

*Finally, related to the dye labeling, I wondered whether the region of overlap indicated by asterisks in Fig 1 would show up as quantitatively similar to the degree of overlap that occurs in the mutants. I believe that that region is not included in any of the sampled areas, but if it is, please note. Additionally, it would be good to comment briefly on how the lack of segregation in this region in control animals might affect the overall model for how *Emx2* acts in developing sensory cells to influence the final connectivity of the VGNs. Yes, the reviewer is correct that the region of dye overlap was not included in our analysis. We, too, were puzzled at the potential function of a region that receives two different types of afferents. Even though the overlap of dye label in the anterior region of the utricle has been described*

previously (Maklad et al, 2010), this is a region where the afferents for the utricle and the anterior and lateral cristae course underneath the sensory epithelium. These afferents could be easily misinterpreted as dual labeling within the sensory epithelium. However, our careful examination of selected z-stack images of the sensory epithelium showed predominant labeling of dye from the cerebellum but also some dye labels from the brainstem within this anterior region. Whether this dual labeling represents functional innervation from both type of afferents is not clear and will require a more in-depth study in the future.

6. Most of the dye labeling is done at E16.5, but it isn't clear why this embryonic timepoint was selected. In general, a bit more information about what is known about VGN innervation patterns and their development could be added to the introduction. On a related note, a lot happens between E16.5 and four months of age, the earliest timepoint for the functional assays. It would be good to examine at least one mature timepoint to see if the pattern seen at E16.5 is stable. In particular, I wondered whether some of the observed lack of segregation could be caused by retention of extra branches that might subsequently be pruned during subsequent synaptic refinement events.

Based on the published results by Bernd Fritsch's laboratory (Maklad et al, 2010), the neuronal segregation pattern is established by E15.5 and remained as such at least at P21. Our wildtype dye tracing results at E16.5 and P0 shown in Figure 6 also supported his results. Thus, the specification of this neuronal patterning is a developmental event and the role of Emx2 in this specification should be investigated early. Additionally, Emx2 KO mice die at birth, which negates postnatal analysis. The developmental timing of this innervation pattern has now been added to the introduction. See P.4, line 79.

Reviewer #2

1. Quantitative analysis of dye tracing data is key in establishing the role of Emx2 in vestibular innervation. However, the statistical analysis used may not be the most appropriate. Specifically, without normality tests, one-way ANOVA is not valid for the small sample sizes analyzed in all the tracing experiments (n=2-6). A non-parametric test that makes no assumption about the distribution of data, such as the Kolmogorov-Smirnov test, may be better at revealing regional innervation differences between different genotypes. This would apply to data presented in Figures 3-6.

Please refer to response #5 to Reviewer #1. We agree with the reviewer. We chose the non-parametric rank-based equivalent to one-way ANOVA, Kruskal-Wallis rank sum test with post-hoc Dunn's pairwise comparisons because we need to compare two or three genotypes between three regions.

2. Several of the Cre lines used are also expressed in brain regions outside of inner ear, which could confound phenotype interpretation, in terms of afferent projection and behavioral defects. For example, Gfi1-Cre is expressed in the cerebellum, and the Cre allele results in Gfi haploinsufficiency. It would be important to use Gfi-Cre/+ samples as "control" groups in experiments involving this Cre driver.

Our preference for the choice of controls is always littermates of the conditional mutants in the following order: 1) *Gfi1cre/+; lox/+*, 2) *lox/-* without the *cre* allele and, 3) *lox/+* alone, which we considered as wildtype. Unfortunately, using *Gfi1cre/+; lox/+* alone as controls does not give us enough sample size for statistical analyses.

Nevertheless, our use of two different cre lines (*Emx2 cre* and *Gfi1 cre*) and two different approaches (changing bundle orientation versus loss of mechanotransduction) to generate the bidirectional mutants, *Emx2 cKO* and *Tmie cKO*, gave us confidence in our conclusions. Also,

Emx2 is not known to be expressed in the cerebellum normally (see response to Reviewer #1, #1), so the cerebellum is not a concern for the interpretation of *Emx2* cKO results.

3. Related to above, the Sox2creER and PIPcreER drivers are expressed in many types of glial cells beside SCs in the sensory epithelia. For Emx2 overexpression experiments, a potential caveat is that Emx2 overexpression in various brain regions affected central projection patterns of vestibular afferents, contributing to the observed dye tracing defects. This is plausible because vestibular afferents are normally sensitive to Emx2-mediated cues from the sensory maculae, and should be at least discussed.

We agree with the reviewer that the gain-of-function models can be difficult to interpret, and they can only serve as supportive evidence for the knockout phenotype. We have included a discussion of this limitation on P13, line 269.

4. Dye tracing results in Figure 6 should be quantified.

Dye-tracing results in Figure 6 have been quantified and those results are presented in Supplementary Figure 5 of the revised manuscript. The assumptions for the linear regression were not satisfied partially due to the relatively small sample size, but Kruskal-Wallis tests provided very strong evidence of a difference ($p < 2.2e-16$) in the relative cb signal between at least one pair of regions. Dunn's pairwise tests were carried out for three pairs of regions for each age group and genotype. While there was significant differences between Lateral and Intermed_Medial or Lateral and Medial regions in each genotype, there was no significant difference in the relative cb signal between controls and mutants at E16.5 (Kruskal-Wallis $H(2) = 1.55$, $p = 0.46$) or P0 (Kruskal-Wallis $H(2) = 1.07$, $p = 0.59$) for any regions.

Reviewer #3

Reviewer #3 (Remarks to the Author):

1. The combination of Emx2 CKO, Tmie CKO and vestibular behaviors demonstrate an overall focus on vestibular circuitry yet the abstract and introduction largely emphasize bundle polarity and bidirectional responses at the level of the hair cells. The authors should consider whether the unique aspects fo their research are overshadowed by this structural organization. One could envision an abstract that began ... "The vestibular maculae are innervated by two parallel afferent projections to the hindbrain and cerebellum which are coordinated with the line of polarity reversal. This innervation is coordinated with bundle polarity but the significance is not known"

We sincerely thank the reviewer for the suggestion, and we have revised the abstract accordingly.

2. Writing styles differ between authors but one might argue that there is an unusually long summary of results in the last paragraph of the introduction

We have revised the last paragraph of the introduction.

3. Line 96: The statement that "vestibular afferents that project to the cerebellum is replaced by neurons that projected to the brainstem in Emx2 knockouts" seems like an overstatement since these fibers still appear present (albeit reduced) in Fig2D.

This sentence was removed from the shortened introduction.

4. Background strains for *Tmie* CKO and *Emx2* CKO mice are not stated. As a result it is unclear whether it is appropriate for vestibular testing to combine control mice from these two different experimental crosses. Both *Tmie* cKO and *Emx2* cKO are both on a mixed CD-1/C57BL6/129 background. This information has been added to P.23, line 493. The controls were littermate controls selected with preference for cre heterozygotes (see response to Reviewer 2, #2 above). The impetus of combining the controls is to be able to compare the relative extent of phenotypes between the two mutants. However, when we split up the two groups of mutants and their littermate controls, the phenotypes on forced swim test and VsEP are the same (see graphs below).

5. Line 168: The following interpretation of *Emx2* KO data “Together, these results suggest that normal *Emx2* expression in the lateral utricle and inner saccule selects for neurons that project to the cerebellum and inhibits neurons that project to the brainstem.” Is inconsistent with *Emx2* overexpression data because neurons projecting to the brain stem are still present and therefore are not subject to ‘inhibition’ We understand the reviewer’s point of view but one could argue that the ectopic *Emx2* in hair cells or supporting cells is not sufficient to mediate full inhibition, unlike the endogenous *Emx2* domain. Nevertheless, we have revised this statement. See P.8, line 157.

6. Similarly in Discussion line 407 “it is equally likely that the exclusion of brainstem neurons is indirectly mediated by neurons that normally project to the cerebellum.” It is hard to rationalize an exclusion model when the manuscript contains experimental conditions in which neurons overlap. Since there are different explanations for the afferent phenotype in *Emx2* gain and loss of function mutant, we decided to refrain from the speculation and simplify the discussion. See P.20, line 409.

7. The data reported as a CB:BS ratio does not appear to be an actual ratio (in which case one would expect values greater than 1 in lateral regions). Instead these appears to report of percentage of green pixels (which would be fine). Minor related point, the methods describes the ratio of green:red whereas the figures are green:magenta

We thank the reviewer for catching this oversight, we have now reported the data as relative CB signals, which is CB/(CB+BS) signals.

8. Lines 259-263 how is it that *Plp1Cre* is not appropriate to generate SC-specific *Emx2* CKOs but is the optimal choice for inducing SC-specific *Emx2* overexpression?

The published paper on *PlpcreER* by Gomez-Casti et al (2010) showed good cre activity in supporting cells of utricular sections. However, when we activated cre activity at E13.5 using tamoxifen, the lateral region of the utricle did not show sufficient reporter activity in wholmount preparations, whereas the medial region was fine (see figure below, n=3). Therefore, the *PlpcreER* strain was only used for ectopic *Emx2* expression and not conditional knockout.

In addition, we also tried using *Gfap-cre* line, which also was not effective based on the reporter expression (see picture below, n= 2). We have clarified this point on P.10, line 215.

We hope you will find our revised manuscript in good order. We look forward to hearing from you!

Best regards,

Doris Wu, Ph.D

REVIEWERS' COMMENTS

Reviewer #1 (Remarks to the Author):

In response to the reviewers' suggestions, the authors have made a number of improvements to the manuscript, including addition of clarifying statements and more appropriate statistical analysis and display of dye labeling data. I am pleased with the final product, which is a compelling demonstration of the circuitry and behavioral impact of the LPR in mice.

Reviewer #2 (Remarks to the Author):

The revised manuscript by Ji et al. has addressed all the concerns I raised previously. Overall, the clarity and statistical rigor of the manuscript has greatly improved, and the findings that regional Emx2 expression in the balance organs regulates afferent projection patterns are convincing and interesting.

A very minor suggestion: for the linear regression analysis, it would be helpful to clarify 1) what is the basis for excluding outliers (except for not fitting a linear model, they could actually reflect inherent variability of the experimental data, e.g., variable Emx2 expression levels following TM induction); 2) what is the interaction term added (p31, line 671), i.e, which variables are interacting.

Reviewer #3 (Remarks to the Author):

the authors have addressed all of my concerns and I leave the final decision in the hands of the managing editor

Response to reviewer #2

*A very minor suggestion: for the linear regression analysis, it would be helpful to clarify 1) what is the basis for excluding outliers (except for not fitting a linear model, they could actually reflect inherent variability of the experimental data, e.g., variable *Emx2* expression levels following TM induction); 2) what is the interaction term added (p31, line 671), i.e, which variables are interacting.*

We thank the reviewer's suggestion. The basis for excluding outliers was two-fold: First, the outlier observations need to lie outside 1.5*IQR (Inter Quartile Range), i.e., the data points outside the whiskers in the box plot in the figure below. (It indicated that these observations were not representative of the intended study population, debatable due to small sample size)

Second, these datapoints are influential outliers identified by large Cook's distance. Cook's distance is a combination of each observations' leverage and residual size. The influential outliers could possess so much power to distort the model. In general, those observations that have a Cook's distance greater than 4 times the mean may be classified as influential.

669 Outliers were identified with the diagnostics plots. After the removal of four outliers in
670 *Sox2^{CreER}; Rosa^{Emx2}* data and three outliers in *Gfi1^{Cre}; Rosa^{Emx2}* and *Plp^{CreER}; Rosa^{Emx2}*
671 data, as well as the addition of an interaction term in the latter data, all the assumptions
672 were satisfied, which include normal distribution and independence of residuals,
673 homoscedasticity, and linear relationship between dependent and independent
674 variables.

Page 31, Line 673-683 (New statement)

Outliers were identified with the diagnostics plots. The outlier exclusion was on two-fold basis:
First, the outliers need to lie outside 1.5*IQR (Inter Quartile Range), as visualized in the box-
whisker plot. Second, the outliers were influential datapoints identified by Cook's distance,
which were at least greater than four times the mean. After the removal of five outliers in
Sox2^{CreER}; Rosa^{Emx2} data and three outliers in *Gfi1^{Cre}; Rosa^{Emx2}* and *Plp^{CreER}; Rosa^{Emx2}* data, as well
as the addition of an interaction term Region*Genotype, all the assumptions were satisfied,
which include normal distribution and independence of residuals, homoscedasticity, and linear
relationship between dependent and independent variables. The model fit has been improved
substantially. We have improved the adjusted R² from 0.45 to 0.59, and from 0.47 to 0.53 for
the above-mentioned data.

The interaction term that was added to the linear regression model for the *Gfi1^{Cre}; Rosa^{Emx2}* and
Plp^{CreER}; Rosa^{Emx2} data was Region*Genotype. The analysis found that the following terms were
significant and with positive coefficients (Supplementary Table 3):

Intermed_Medial : *Gfi^{Cre}; Rosa^{Emx2}*

Intermed_Medial : *Plp^{CreER}; Rosa^{Emx2}*

Medial : *Gfi^{Cre}; Rosa^{Emx2}*

Medial : *Plp^{CreER}; Rosa^{Emx2}*

It indicated that the increase of the cb signal in *Gfi^{cre}* and *Plp^{creER}* strains compared to controls
only occurred in the intermed_medial and medial regions, but not in lateral region.